# Towards Better Generalization with Flexible Representation of Multi-Module Graph Neural Networks

**Hyungeun Lee**                                                                *didls1228@hanyang.ac.kr*
*Department of Electronic Engineering*
*Hanyang University*

**Kijung Yoon**                                                                  *kiyoon@hanyang.ac.kr*
*Department of Electronic Engineering*
*Hanyang University*

**Reviewed on OpenReview:** *https://openreview.net/forum?id=EYjfLeJL4l*

## Abstract

Graph neural networks (GNNs) have become compelling models designed to perform learning and inference on graph-structured data. However, little work has been done to understand the fundamental limitations of GNNs for scaling to larger graphs and generalizing to out-of-distribution (OOD) inputs. In this paper, we use a random graph generator to systematically investigate how the graph size and structural properties affect the predictive performance of GNNs. We present specific evidence that the average node degree is a key feature in determining whether GNNs can generalize to unseen graphs, and that the use of multiple node update functions can improve the generalization performance of GNNs when dealing with graphs of multimodal degree distributions. Accordingly, we propose a multi-module GNN framework that allows the network to adapt flexibly to new graphs by generalizing a single canonical nonlinear transformation over aggregated inputs. Our results show that the multi-module GNNs improve the OOD generalization on a variety of inference tasks in the direction of diverse structural features.

## 1 Introduction

Graph neural networks (GNNs) have recently been established as an effective machine learning models for representation learning on graph-structured data (Scarselli et al., 2008; Hamilton et al., 2017b; Bronstein et al., 2017; Battaglia et al., 2018). Graph is a powerful mathematical abstraction that can represent the structure of many complex data, and learning on graphs has been widely explored in many scientific domains (Duvenaud et al., 2015; Gilmer et al., 2017; Sanchez-Gonzalez et al., 2018; 2020; Gainza et al., 2020).

Despite the growing empirical success of GNNs in various fields, the necessity of a deeper understanding of this model framework has arisen due to the inconsistent model efficiency across different tasks and experimental settings. Building large-scale graph benchmarks (Dwivedi et al., 2020; Hu et al., 2020) is one recent attempt to address this challenge, and there have also been significant theoretical studies on the expressivity of GNNs focused on the isomorphism task (Xu et al., 2019; Morris et al., 2019; Garg et al., 2020; Sato, 2020; Morris et al., 2021). However, little work has been conducted on understanding the fundamental limitations of GNNs for adapting to distribution shifts on graphs, where systematic differences between training and test data can significantly degrade model performance.

Recent approaches to OOD generalization (Ovadia et al., 2019; Arjovsky et al., 2019; Ahuja et al., 2021; Lu et al., 2021) concentrate primarily on images or structural equation models. However, the nature of the graph domain is fundamentally different from these works in that the inputs are not simple image features or variables, but full of complex irregularities and connectivity in topology. We hypothesize that the difference in underlying graph properties of training and testing datasets (i.e. structural distribution shift) presents

a fundamental challenge to the extrapolation beyond the range of input distribution. We are particularly interested in the conditions under which we can expect GNNs to (not) perform well in predicting targets for unseen graphs. To study this question, we use a random graph generator that allows us to systematically investigate how the graph size and structural properties affect the predictive performance of GNNs.

We argue that, among the many graph properties, the average node degree within each mode of the degree distribution is a key factor that determine whether GNNs can generalize to unseen graphs. Then, we propose and explore methods for using multiple node update functions as a way to generalize a single canonical nonlinear transformation over aggregated inputs. This approach enhances the flexibility of the network to handle shifts in graph properties, resulting in better control over structural distribution shifts. We evaluate the performance of the multi-module GNN framework on the task of approximating marginal inference for general graphs (Pearl, 1988), solving graph theory multi-task problems (Corso et al., 2020), conducting CLRS algorithmic reasoning tasks (Veličković et al., 2022), and benchmarking against OOD scenarios for real-world graphs (Gui et al., 2022). Our key contributions are as follows:

1. In Section 3.1, we present a diverse and densely populated random graph benchmark that enables the identification of visually recognizable and spatially characterized generalization patterns of GNNs. The random graphs offered in this study can be utilized as a useful testbed for graph OOD benchmark, as they allow for the integration of task-specific node/edge/graph-level input features and target outputs.

2. In Section 5.1, we identify a strong generalization pattern in OOD scenarios that is strongly correlated with the average node degree of in-distribution (ID) training graphs. This pattern remains consistent across various graph sizes, tasks, and baseline GNN models.

3. In Section 5.2, we demonstrate that having multiple modes in the degree distribution can impede OOD generalization, as it alters the distribution of incoming messages. Therefore, when evaluating generalization on unknown graphs, we suggest that it is important to consider the average node degree for each mode of the degree distribution separately instead of evaluating the distribution as a whole.

4. In Section 4, we propose a multi-module GNN framework designed to leverage multiple update modules to process complex messages aggregated from nodes in far modes of the degree distribution, leading to improved OOD generalization for several inference tasks (Section 5.3).

## 2 Preliminaries

The defining feature of GNNs is the form of neural message passing where vector messages are exchanged between nodes and updated on arbitrary graph structure using neural networks (Gilmer et al., 2017). The message passing operation that underlies variants of GNNs follows a common strategy: at each iteration, every node computes incoming MESSAGEs ($\mathcal{M}$) from its local neighborhood, AGGREGATEs ($\bigoplus$) them, and UPDATEs ($\mathcal{U}$) its node state $\mathbf{h}_i$ by

$$\mathbf{h}_i^{(t+1)} = \mathcal{U}\left(\mathbf{h}_i^{(t)}, \bigoplus_{(j,i) \in \mathcal{E}} \mathcal{M}\left(\mathbf{h}_i^{(t)}, \mathbf{h}_j^{(t)}, \mathbf{z}_{i,j}\right)\right) \tag{1}$$

where $\mathbf{z}_{i,j}$ represents the feature embedding vectors for input nodes $i, j$ in the node set $\mathcal{V}$ and for the edge $(j,i)$ in the edge set $\mathcal{E}$, and $\mathcal{M}$ and $\mathcal{U}$ are arbitrary differentiable functions (i.e., neural networks). The iterative scheme relies on reusing and sharing the three main mathematical operations (i.e., MESSAGE, AGGREGATE, and UPDATE) across computational graphs, where functional modules can operate on graphs of varied sizes and structures. Such a flexible design principle allows GNNs to be related to standard neural network layers (Battaglia et al., 2018).

As an illustrative example, multilayer perceptron (MLP) layers, where each node takes in the sum (AGGREGATE) of weighted inputs (MESSAGEs) from a previous layer and passes (UPDATEs) it through a fixed nonlinear

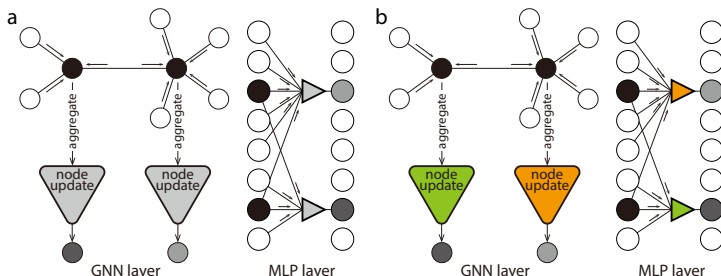

Figure 1: **Overview of the proposed model structures. (a)** GNN layer (left) and MLP layer (right) with a fixed nonlinear update/activation function (gray triangles). **(b)** The same layers with two different nonlinearities (green and orange triangles).

activation function, are comparable to GNN layers (Figure 1a), although the all-to-all relations between MLP layers are different from the GNN relations defined by the edges. As conceptual extensions from scalar to vector-valued inputs, a shared UPDATE function in GNNs can be viewed as the same elementary building block as a canonical activation function such as ReLU in MLPs, in the sense that they all perform nonlinear transformations over aggregated inputs (Figure 1a). From this point of view, the proposed idea of using multiple update functions in GNNs is analogous to the use of multiple activations in MLPs (Figure 1b).

The motivation behind the use of multiple nonlinearities is to simulate the effect of diverse nonlinear cell properties in the brain (Douglas & Martin, 1991; Shepherd, 2004). Recent work has demonstrated that the task-dependent relative proportion of two distinct cell types allows sample-efficient learning and improves the generalization performance on orientation discrimination tasks (Bordelon & Pehlevan, 2022). The fact that an optimal mixture of two different cell types exists suggests that more than one nonlinear function type might be required for better generalization. The benefits for machine learning at this level of collective nonlinear input-output relationships remain unclear, and to the best of our knowledge, this is the first study to characterize the impact of multi-module nonlinear functions and their usefulness in the generalization of GNNs.

## 3    Experimental Setup

In this section, we present our approach for generating random graphs and analyzing their graph properties through visualization. Then we detail how these random graphs will be applied to specific graph benchmarks.

### 3.1    Random graph benchmark

**Random graph generator**. Our goal is to create a random graph generator that can produce a diverse set of graphs with a wide range of graph properties. Building on previous work (You et al., 2020), we have relaxed the constraint of the Watts-Strogatz model to allow for different node degrees before random rewiring. To generate a graph with $n$ nodes and an average node degree of $k$, we first set the number of edges to $e = \lfloor \frac{nk}{2} \rfloor$. Then, we construct a ring graph in which each node is connected to its $\lfloor \frac{e}{n} \rfloor$ subsequent neighbors. After that, we randomly select $e \bmod n$ nodes and connect them to their nearest unconnected neighbors. Finally, we randomly rewire all edges with probability $p$. By repeating this process with different random seeds for each parameter setting, we have generated a total of 3,600,000 random graphs (more information is provided in Appendix A.1). This method, although it appears to be a single model, can generate a variety of graphs that encompass those produced by classic random graph generators like Erdős-Rényi (Erdos et al., 1960), Watts-Strogatz (Watts & Strogatz, 1998), Barabási-Albert (Albert & Barabási, 2002), and Harary (Harary, 1962) models, including ring and complete graphs (Figure A1).

**Visualizing structural properties of random graphs**. We next compute six graph properties for all samples. These include the average shortest distance between nodes (*average path length*), the fraction of connected triangles in a graph (*clustering coefficient*), the *average, maximum, minimum*, and *standard*

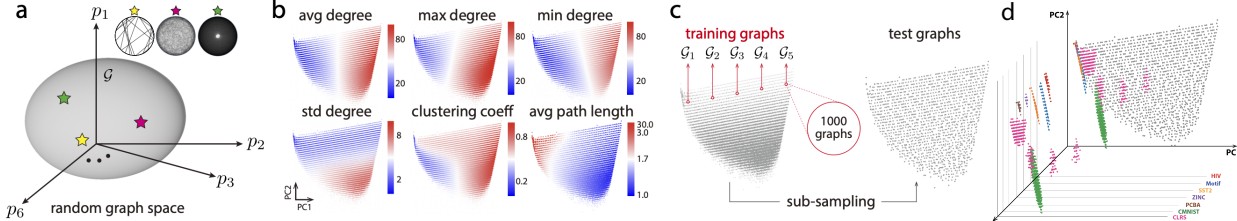

Figure 2: **Random and real-world graph benchmarks. (a)** Illustration of random graph space where each axis $p_i$ represents a graph measure. Inset: Three examples of 100-node graphs (stars) with different connectivity levels. **(b)** 2D scatter plots showing the distribution of each of the six graph properties across the entire dataset (N=3.6M graphs). **(c)** Left: The graphs (gray dots) in red circles constitute five distinct sets $\mathcal{G}_{i=1:5}$ of training graphs, Right: Uniformly sub-sampled test graphs (gray dots). **(d)** Seven distinct real-world graph benchmarks are displayed with different colors, and they are superimposed on each other. To avoid confusion arising from overlapping distributions, each dataset is independently re-visualized along the orthogonal direction in 2D space.

*deviation* of node degrees. Each graph is then characterized by six measures, and the value of these variables can be naturally mapped to a point (vector) in a six-dimensional (6D) space where each axis represents one of the variables (Figure 2a). If there are a large number of graphs and the points are close to one another, they can form a cluster with distinct components in this space (Figure 2a). To better understand the distribution of sampled graph properties, we use principal component analysis to reduce the 6D space to 2D and visualize them by presenting the recorded measures for each sampled graph as color-coded scatter plots (Figure 2b). The visualized results demonstrate the development of the structural properties of the sampled graphs in a low-dimensional space (Figure 2b). The progression in these patterns will be used as a means for evaluating the generalization capabilities of GNNs.

**Data split for OOD tasks**. This set of large random graphs serves as a test bed for assessing the generalization performance of proposed models in OOD settings. Specifically, we choose five evenly spaced locations on a 2D scatter plot, as shown in Figure 2c. We then collect all samples within a circle of a half-unit radius centered on each position and consider only non-isomorphic graphs. The resulting set $\mathcal{G}_i$ of non-isomorphic graphs obtained from the $i$-th location composes a distinct set of training graphs (Figure 2c). The number of training samples in each group may differ slightly; thus, we ensure that the sample count is consistently 1,000 across the five groups $\mathcal{G}_{i=1:5}$ by providing different input features to the graphs. For the test graphs $\mathcal{G}_{\text{test}}$, we uniformly sub-sample $\sim$1,000 graphs from the entire sample space ($N$=3.6M) as in Figure 2c (more details in Appendix A.1). Since the training split occupies a small portion of the sample space compared to the test split, this data split is appropriate for studying OOD generalization in the presence of structure shift. In our experiments for the presented benchmark, we use graphs with small to medium sizes ranging from 16 to 100 nodes.

## 3.2 Specific tasks for input features and targets

Our random graph dataset can be used for various tasks in graph representation learning simply by assigning node and edge features with corresponding targets to the predefined graph structures. We first consider two specific tasks in this work: (1) an approximate marginal inference task that is intractable for general graphs and (2) a multi-task graph theory problem consisting of three node-level and three graph-level prediction tasks. Since these two tasks are defined on the previously created random graphs, we manually generate synthetic input features and target outputs for each task.

**Marginal inference task**. Computing marginal probability distributions of task-relevant variables at each node of a graph is often computationally intractable, and thus relies on approximate methods (Pearl, 1988; Wainwright et al., 2003). We follow the setup of recent work improving message-passing algorithms by GNNs (Yoon et al., 2018; Satorras & Welling, 2021). Specifically, we evaluate the generalization performance in Markov random fields (Ising model) comprising binary variables $\mathbf{x} \in \{+1, -1\}^{|\mathcal{V}|}$, each with an associated

bias $b_i$ and coupling strength $J_{ij}$ via the following parameterization:

$$p(\mathbf{x}; \boldsymbol{\theta}) = \frac{1}{Z(\boldsymbol{\theta})} \exp \left( \sum_{i \in \mathcal{V}} b_i x_i + \sum_{(i,j) \in \mathcal{E}} J_{ij} x_i x_j \right) \tag{2}$$

The collective model parameters $\boldsymbol{\theta} = \{b_i\}_{i \in \mathcal{V}} \cup \{J_{ij}\}_{(i,j) \in \mathcal{E}}$ determine the state of each variable node, given the state of other connected nodes. The number of nodes $|\mathcal{V}|$ and the connectivity structure $\mathcal{E}$ are determined by our random graph benchmark. The Ising model parameter $\boldsymbol{\theta}$ is sampled from a standard normal distribution $\mathcal{N}(0, 1)$, where the sampled bias $\{b_i\}$ and coupling weight $\{J_{ij}\}$ are provided as input node and edge features of GNNs, respectively. For the sake of the targets, we calculate marginal probabilities $p(x_i)$ for each node $i \in \mathcal{V}$ of small graphs ($|\mathcal{V}| = 16$) using brute-force enumeration, whereas we run a Markov Chain Monte Carlo (MCMC) algorithm, the Gibbs sampler (Geman & Geman, 1984), on larger graphs ($|\mathcal{V}| = 36, 100$). We accept the approximate marginals as ground truth targets when the mean absolute error of the estimates from the Gibbs sampler with respect to the counterpart of the binary Hamiltonian Monte Carlo (Pakman & Paninski, 2013) falls below 0.02, suggesting that convergence is reached for the MCMC chains: $\langle |p_{\text{Gibbs}}(x_i) - p_{\text{HMC}}(x_i)| \rangle_{i \in \mathcal{V}} \leq 0.02$.

**Graph theory multi-task**. We also investigate classical graph theory problems at both the node and graph levels. At the node level, we focus on predicting single-source shortest path lengths, eccentricity, and Laplacian features[1], while at the graph level, we aim to determine diameter, spectral radius, and whether the graph is connected. These tasks were initially addressed by Corso et al. (2020), but in this work, we evaluate them on 100-node random graphs that are more diverse in structure and larger in size than in previous studies. As input features, the graph is provided with two types of position indices for each node: a one-hot vector representing the source for the shortest path task and a random scalar uniformly sampled from the range of [0,1].

### 3.3 Real-world graph benchmark

In order to explore the practical implications of our proposed model framework beyond the random graph benchmark, we have also considered real graph benchmark datasets and visualized their graph properties in the 2D scatter plot (Figure 2d). Our aim is to gain a better understanding of how real-world graphs are distributed and assess whether they accurately represent the full range of graph properties. Interestingly, we discover that most of these datasets have limitations either by sparsely populating or occupying only a small portion of the graph space (Figure 2d). This supports the idea that our random graph benchmark is beneficial for studying structural shifts in distribution, rather than solely relying on real-world scenarios. Nevertheless, we intend to experiment with the proposed ideas using real-world datasets and then evaluate the outcomes by comparing them with our observations from studying the random graph benchmark. Ultimately, this will enable us to validate the practical utility of our approach.

**CLRS algorithmic reasoning tasks**. The CLRS benchmark is a recent comprehensive dataset that includes 30 highly complex classical algorithmic reasoning tasks (Veličković et al., 2022). Although most algorithms in CLRS are based on smaller graphs[2] generated from a single Erdős-Rényi model, we have opted to evaluate our proposed methods using CLRS because it is a leading benchmark for investigating OOD generalization in graph domains. The summary of dataset configuration is provided in Table A3.

**GOOD: Graph OOD benchmark**. GOOD (Gui et al., 2022) is the latest real-world graph OOD benchmark designed to be challenging by making distinct shifts in input or target distribution (i.e. covariate and concept shifts). It provides 6 graph-level and 5 node-level prediction tasks, covering a range of datasets from molecular graphs and citation networks to colored MNIST datasets. Our focus is on graph prediction tasks exclusively (more details in Table A4), as the node-level tasks are all based on a single graph (i.e., transductive setting), which falls outside the scope of our study that aims to explore generalization to unseen graphs across multiple graphs.

---

[1] i.e., $LX$ where $L = D - A$ is the Laplacian matrix and $X$ is the node feature vector.
[2] The size of the training and validation graphs is 16 nodes, while the test graphs have 64 nodes.

# 4 Multi-module GNNs

We next present various strategies for building multi-module GNNs. The GNN architecture that we examine utilizes an encode-process-decode framework (Battaglia et al., 2018) as depicted in Figure 3a, and the baseline processor is based on the message-passing neural network (MPNN; Gilmer et al., 2017), as outlined in Equation 1. Additionally, we incorporate the graph attention mechanism (GAT; Veličković et al., 2018) when appropriate to improve generalization performance. The update function in the processor is a crucial component of the proposed methods and plays a vital role in the performance of GNNs. Thus, we outline three different strategies for leveraging two update modules within our framework[3].

## 4.1 Multi-module sigmoid gating

The first approach is to use a sigmoid gating mechanism (Figure 3b, left), where the values of the learned gates determine the utilization of multiple update modules. Specifically, the next hidden state $\mathbf{h}_i^{(t+1)}$ is computed by taking a linear combination of two intermediate node states $\mathbf{h}_{i,1}^{(t)}$ and $\mathbf{h}_{i,2}^{(t)}$ as follows:

$$\mathbf{h}_i^{(t+1)} \quad = \quad \alpha_i^{(t)}\mathbf{h}_{i,1}^{(t)} + (1 - \alpha_i^{(t)})\mathbf{h}_{i,2}^{(t)} \tag{3}$$

where the two intermediate representations $\mathbf{h}_{i,1}^{(t)}$ and $\mathbf{h}_{i,2}^{(t)}$ are the outputs of Equation 1 using distinct update functions $\mathcal{U}_1$ and $\mathcal{U}_2$, respectively. The value of the gating variable $\alpha_i^{(t)}$ is obtained from another processor but replacing $\mathcal{U}$ with a gating function $\phi_g$, which is a neural network with a sigmoid activation for the scalar output.

## 4.2 Multi-module binary gating

The second approach for updating the node states in our model involves using a binary gating mechanism (Figure 3b, middle), where a binary decision is made between two different update modules using the Gumbel reparameterization trick (Jang et al., 2017; Maddison et al., 2017). Unlike the sigmoid gating method, this approach allows to only applies one specific update module to each node. To accomplish this, we set $\alpha_i^{(t)}$ as a random variable with a Bernoulli distribution, parameterized by $\pi_i^{(t)} \in [0, 1]$, such that $\alpha_i^{(t)} \sim \text{Ber}(\pi_i^{(t)})$. However, this parameterization poses a challenge in terms of differentiability, as the gradient does not flow through $\alpha_i^{(t)}$ in a typical Bernoulli sampling. To overcome this issue, we apply the Gumbel reparameterization trick as follows:

$$\alpha_i^{(t)} = \sigma \left( \frac{\log(\pi_i^{(t)}/(1 - \pi_i^{(t)})) + (g_{i,1}^{(t)} - g_{i,2}^{(t)})}{\tau} \right) \tag{4}$$

where $g_{i,1}^{(t)}, g_{i,2}^{(t)} \sim \text{Gumbel}(0, 1)$, $\pi_i^{(t)}$ is the output of $\phi_g$, and $\alpha_i^{(t)} = 1$ with probability $\pi_i^{(t)}$ as the temperature $\tau$ approaches 0. In this way, the next node state $\mathbf{h}_i^{(t+1)}$ becomes one of two intermediate representations $\mathbf{h}_{i,1}^{(t)}$ and $\mathbf{h}_{i,2}^{(t)}$. The binary gating strategy can be considered as sitting in the middle of the spectrum among the three strategies we introduce. During the forward pass, binary gating employs discrete sampling similar to the approach described next in multi-module meta-learning, while during the backward pass, computations are carried out on a computational graph defined by multiple modules, akin to sigmoid gating.

## 4.3 Multi-module meta-learning

The last strategy is to learn a set of update modules that can be inserted in various combinations at different nodes within the processor (Figure 3b, right). This method is significantly different from the previous gating mechanism as the selection of the modules and the updating of their parameters take place at different phases through meta-learning (Schmidhuber, 1987; Thrun & Pratt, 2012). In particular, the goal is to distribute two

---

[3]It should be noted that the methods described for two modules can be easily generalized to multiple modules greater than two. For a comprehensive account of the model and optimization hyperparameters, please refer to Appendix A.2.

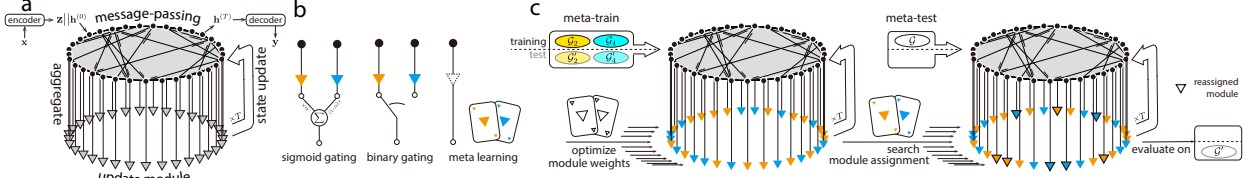

Figure 3: **Multi-module GNNs. (a)** Illustration of an encode-process-decode framework with the emphasis on the processor as a message-passing neural network. Every node (black dot) computes incoming messages (black arrows), aggregate (vertical line) them, and updates (triangle) its node state. **(b)** Three different ways of leveraging two update modules (orange and blue triangles): sigmoid gating (left), binary gating (middle), and multi-module meta-learning (right). **(c)** The multi-module GNN meta-learns a set of update modules, and at meta-test time, the system only needs to determine which modules should be replaced to make the most accurate predictions based on the observed data.

update modules across all nodes of a graph and search over the set of possible structures to find the one that best generalizes in OOD settings. The idea behind this composition is to flexibly reuse and recombine a finite number of update modules to ultimately achieve combinatorial generalization (Tenenbaum et al., 2011; Andreas et al., 2016; Ellis et al., 2018; Alet et al., 2018).

To be specific, let $\Theta = (\boldsymbol{\theta}_1, \boldsymbol{\theta}_2, \boldsymbol{\theta}_{\text{rest}})$ be the weights of update modules $\mathcal{U}_1, \mathcal{U}_2$, and the rest of a multi-module GNN. During meta-training, an optimal assignment $\mathcal{A}$ of two modules is determined for each node in training graphs $\mathcal{G}_i^{\text{train}}$, given a fixed set of model parameters $\Theta$. We then update $\Theta$ using test graphs $\mathcal{G}_i^{\text{test}}$, while maintaining the previously determined module allocation $\mathcal{A}$ for each graph (Figure 3c). This is achieved by using the BounceGrad algorithm (Alet et al., 2018), which alternates between steps of simulated annealing for improving $\mathcal{A}$ and gradient descent for optimizing $\Theta$ to minimize the predictive loss $\mathcal{L}$:

$$\mathcal{L}(\Theta) = \sum_i \ell \left( \mathcal{G}_i^{\text{test}}, \underset{\mathcal{A}_i \in \mathbb{A}}{\arg\min} \, \ell \left( \mathcal{G}_i^{\text{train}}, \mathcal{A}_i, \Theta \right), \Theta \right) \tag{5}$$

In meta-testing phase, we assume the presence of a small number of final training graphs, $\mathcal{G}_{\text{meta-test}}^{\text{train}}$. These are not OOD samples, but rather a limited number of ID graphs. In general, meta-learning algorithms do not share the same task between the meta-training and meta-testing datasets. However, in our approach, we define a task as finding an optimal module assignment for each individual graph in the meta-training phase, while a new task in the meta-testing phase is defined as finding a single module structure that is applicable to all graphs in $\mathcal{G}_{\text{meta-test}}^{\text{train}}$. The objective is to search for a final optimal assignment of update modules over all possible arrangements $\mathbb{A}$ given $\mathcal{G}_{\text{meta-test}}^{\text{train}}$ with fixed $\Theta$:

$$\mathcal{A}^* = \underset{\mathcal{A} \in \mathbb{A}}{\arg\min} \, \ell \left( \mathcal{G}_{\text{meta-test}}^{\text{train}}, \mathcal{A}, \Theta \right) \tag{6}$$

The generalization performance is then evaluated on OOD test split, using the optimal structure $\mathcal{A}^*$. It is important to note that, in contrast to the gating mechanism, the type of update function selected for each node is no longer influenced by the test dataset at the final evaluation.

## 5  Results

We now presents a series of controlled experiments on the random graph benchmark to evaluate the generalization capability of GNNs. Our main objective is to pinpoint key factors that can consistently determine the ability of GNNs to generalize well on OOD inference tasks. We then proceed to examine how the incorporation of multiple update modules within a general GNN framework can enhance their OOD generalization across these tasks.

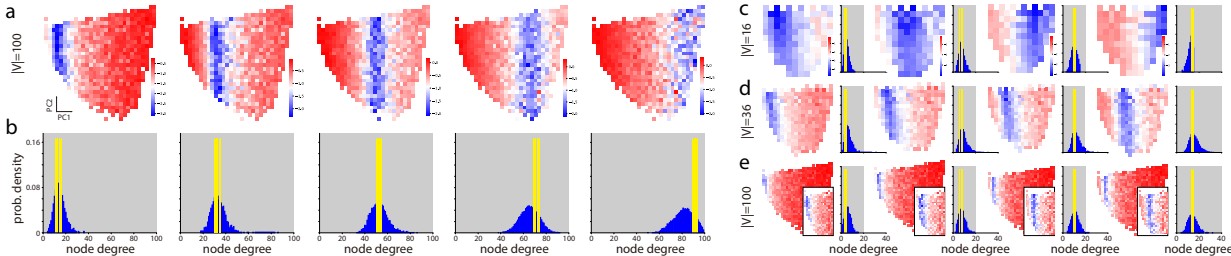

Figure 4: **Structural distribution shift and size generalization (a)** Generalization performance on a 2D subspace as training graphs change from $\mathcal{G}_1$ to $\mathcal{G}_5$. Blue indicates a lower KL divergence (more accurate prediction), while red denotes a higher KL divergence (less accurate prediction). **(b)** Normalized node degree histograms generated from the training graphs (yellow) and the test graphs with the top 30% predictive performance (blue). **(c)** Same as Figure 4a except that the model is trained and evaluated on smaller graphs with $|\mathcal{V}| = 16$. **(d-e)** Testing on larger graphs, $|\mathcal{V}| = 36$ and $|\mathcal{V}| = 100$, respectively. The inset in (e) depicts the magnified region of interest.

## 5.1 Degree matters in OOD generalization

To gain insight into the predictive performance of GNNs, we first consider the marginal inference task. In this task, the models are trained on each group $\mathcal{G}_i$ with a size of $|\mathcal{V}| = 100$, and are tested on $\mathcal{G}_{\text{test}}$ of the same size but with different structures. We use the Kullback-Leibler (KL) divergence as a validation metric and present the results in a heat map (Figure 4a). To do this, we down-sample and aggregate the test graphs in Figure 2c into a coarse resolution of 900 (=30×30) 2D spatial bins, where each bin records the average KL divergence of the graphs within it. For display purpose, we compute the average values in log scale and present them with contrasting colors, with blue indicating a lower KL divergence (or more accurate predicted marginals) and red indicating the opposite. If GNNs only learned representations of the training graphs, we would expect to see a small blue region around the training set $\mathcal{G}_i$. However, all experiments produce a non-trivial vertical band-like pattern translated by the location of $\mathcal{G}_i$ (Figure 4a). It is noteworthy that this vertical pattern aligns with the distribution of similar average node degrees in 2D scatter plot shown in Figure 2b.

To further investigate the role of average node degree in the generalization pattern of GNNs, we generate a normalized histogram of the degrees of every node in the test graphs with the top 30% predictive performance (Figure 4b, blue) and of all training graphs (Figure 4b, yellow). The histograms should have the same peak location if the band-like pattern results from the average degree of training graphs. We observe that the histogram from the testing graphs is unimodal, and the peak location closely coincides with the counterpart of the training graphs (Figure 4b), indicating that the *average node degree* is a critical feature in predicting GNNs' generalization capabilities to unseen graphs. It is worth noting that this generalization pattern is not a feature only seen in the marginal inference task, but is consistently observed across other tasks and baseline GNN models (Figure A2). This suggests that the pattern is not influenced by the specific task and type of GNN architecture employed.

We next examine the size generalization problem by keeping the size of the training graphs small, with $|\mathcal{V}| = 16$, and gradually increasing the size of the test graphs to $|\mathcal{V}| \in \{16, 36, 100\}$. To achieve this, we generate a new set of 3.6M random graphs with a size of $|\mathcal{V}| = 16$, create four sets of training graphs $\mathcal{G}_{i=1:4}$, sub-sample $\mathcal{G}_{\text{test}}$, and perform the same marginal inference task as previously described. If the average degree were still a crucial factor, we would expect to see a band-like pattern that is not affected by the size of the training and testing graphs. Additionally, the corresponding histograms of node degrees should also appear superimposed. Indeed, we find that GNNs trained on $\mathcal{G}_i$ with a size of $|\mathcal{V}| = 16$ produce a pattern like a vertical band aligned with the center of $\mathcal{G}_i$, and the normalized histogram from $\mathcal{G}_i$ provides excellent matches to the peak of the histogram obtained from $\mathcal{G}_{\text{test}}$ consistently across the same size (Figure 4c) and larger (Figure 4d-e) test graphs. This result suggests that the graph size may *not* be a fundamental limiting factor

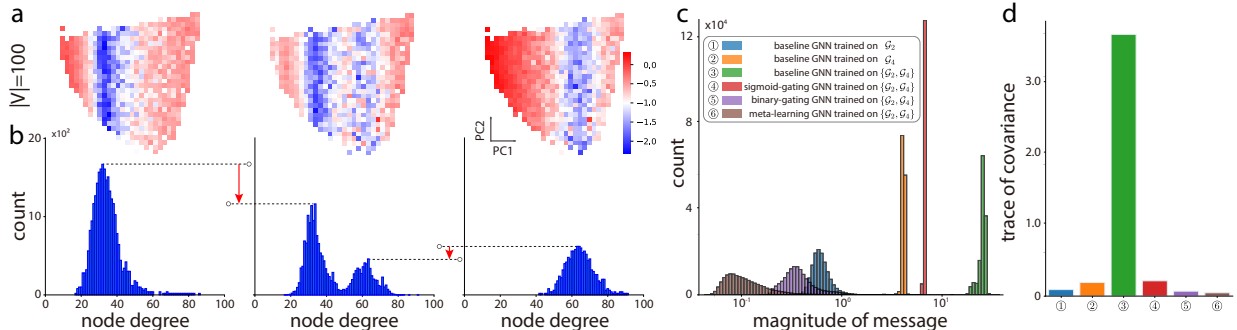

Figure 5: **Effect of bimodal degree distribution on the predictive performance.** **(a)** OOD performance of GNNs when trained on $\mathcal{G}_2$ (left), $\mathcal{G}_2 \& \mathcal{G}_4$ (middle), and $\mathcal{G}_4$ (right). **(b)** Node degree histograms. The number of well-predicted graphs decreases substantially (red arrows) when jointly trained on $\mathcal{G}_2 \& \mathcal{G}_4$. **(c)** The magnitude and **(c)** the total variance of message vectors gathered from both the baseline and multi-module GNNs.

for the generalization in GNNs when the average node degree remains invariant across the training and testing distributions.

## 5.2 Multimode degree distribution as an impediment to OOD generalization

In the previous section, we observe that the degree distribution of graphs on which GNNs generalize well is characterized by a single mode, due to the relatively uniform degrees of the training graphs. However, it is still unclear if the average degree continues to be a significant factor in generalization when training on graphs with highly distinct node degrees. To address this, we create a new training split $\mathcal{G}_{train}$ by combining two existing sets of 100-node graphs, $\mathcal{G}_2$ and $\mathcal{G}_4$, resulting in a sample size ($N$=2000) twice larger than before.

In Figure 5a, the test performance of GNNs trained on $\mathcal{G}_{train}$ is compared to those trained on $\mathcal{G}_2$ and $\mathcal{G}_4$, respectively. The generalization pattern now becomes dual band-like, with a separation determined by the gap between $\mathcal{G}_2$ and $\mathcal{G}_4$ (Figure 5a, middle). This is also reflected in the bimodal appearance of the degree histogram (Figure 5b, middle). These results indicate that the average node degree is not the sole determining factor for generalization, but rather it plays a role in each mode of the degree distribution when evaluating OOD generalization for unknown graphs. Notably, the overall predictive performance for graphs within the pattern decreases significantly when trained simultaneously on $\mathcal{G}_2$ and $\mathcal{G}_4$ compared to when trained individually (Figure 5a-b). This finding suggests that even within a fixed set of graphs, the change in graph composition from a single-mode to multi-mode degree distribution hinders the OOD generalization of GNNs.

We speculate that the degraded performance might be due to the change in the statistics of aggregated messages from nodes with multi-mode degree distributions. To test this hypothesis, an additional experiment is performed where the aggregated message is initially collected from each node in a graph from the GNN models trained on $\mathcal{G}_2$, $\mathcal{G}_4$, and $\mathcal{G}_{train}$, respectively. This procedure is then repeated for all test graphs to acquire a large message sample. We then compute (1) the magnitude and (2) the spread of messages by computing the trace of the covariance matrix of messages (Figure 5c-d). The results reveal that the magnitude of messages significantly increases, and the trace of covariance, which is the total variance of messages, becomes much larger when the baseline model is trained on $\mathcal{G}_2$ and $\mathcal{G}_4$ simultaneously compared to when it is trained separately (Figure 5c-d and Figure A3). The findings indicate that a switch from a unimodal to a multimodal degree distribution leads to a corresponding shift in the summary statistics of the message distribution, which may have created difficulties in the training of GNNs. This situation is similar to the one where several techniques for network initialization (Glorot & Bengio, 2010; He et al., 2015) or normalization (Ioffe & Szegedy, 2015; Ba et al., 2016) have been developed to maintain the mean or variance of activations during the training of deep neural networks.

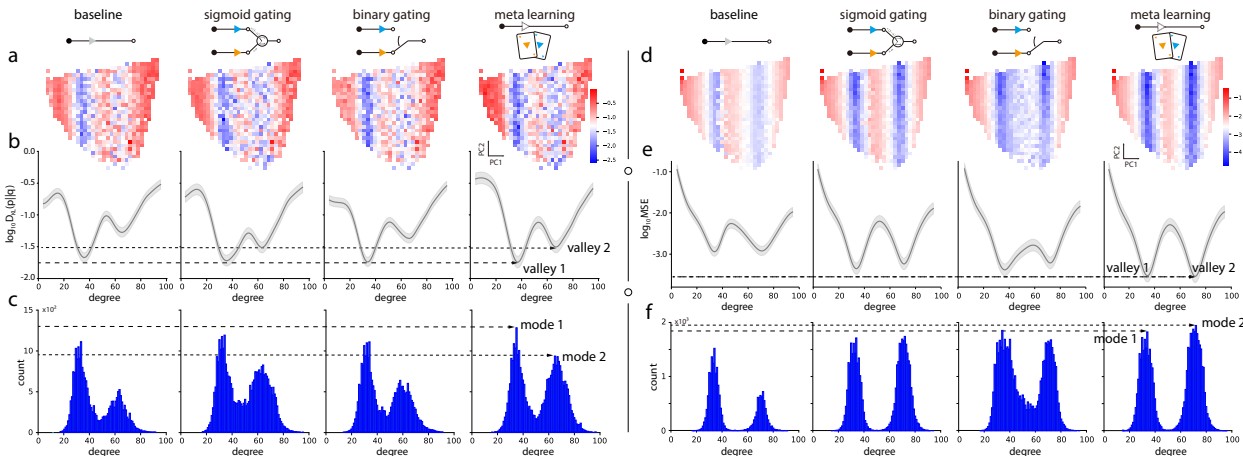

Figure 6: **Performance of different multi-module GNN strategies** on **(a-c)** marginal inference task and **(d-f)** graph theory multi-task. Each column is arranged in the following order: a single module baseline, and multi-module GNNs using sigmoid gating, binary gating, and meta-learning. **(a,d)** Test performance of GNNs when trained on $\mathcal{G}_2$&$\mathcal{G}_4$. **(b,e)** Test KL-divergence loss in log scale (mean $\pm$ sd) **(c,f)** Node degree histogram from the test graphs with the top 40% predictive performance. For ease of comparison, the peak/bottom of each mode/valley in the rightmost column (best model) is highlighted by dashed lines.

Table 1: Test performance of single-module baseline and three different multi-module GNN strategies over 2 benchmark tasks.

| EVALUATION METRIC METHOD | BASELINE | SIGMOID GATING | BINARY GATING | META LEARNING |
|---|---|---|---|---|
| MARGINAL INFERENCE LOSS @ (VALLEY 1, VALLEY 2) # OF NODES @ (MODE 1, MODE 2) | (-1.68, -1.28) (1480, 698) | (-1.73, **-1.52**) (1598, 1099) | (**-1.75**, -1.38) (1477, 848) | (-1.74, -1.51) (**1707, 1245**) |
| GRAPH THEORY MULTI-TASK LOSS @ (VALLEY 1, VALLEY 2) # OF NODES @ (MODE 1, MODE 2) | (-2.94, -2.93) (1522, 723) | (-3.36, -3.24) (1714, 1742) | (-3.38, -3.22) (**1853**, 1679) | (**-3.53, -3.54**) (1825, **1945**) |

## 5.3 Multi-module GNN framework for improving OOD generalization

Our multi-module training framework is proposed to address the issue and achieve improved generalization performance. The framework provides complementary update functions for processing those messages and exploring a diverse set of module combinations during training, which could potentially benefit from an ensemble effect in complex scenarios where the distribution of aggregated messages becomes more intricate. The message analysis applied to this new model framework shows that incorporating multiple update modules in a multimodal degree scenario reduces both the magnitude and variance of incoming messages. This reduction remains significant in some cases, even when compared to the single-module baseline trained on $\mathcal{G}_2$ or $\mathcal{G}_4$ (Figure 5c-d). The results from the previous section and the current analysis lead us to consider these training and test splits as a favorable OOD scenario, as it provides ample opportunity for improving OOD performance. Thus, we aim to address how multi-module GNNs are generalized to this OOD setting for the problems of marginal inference and graph theory multi-task, and show that the multi-module framework brings performance gains over the single-module baseline.

**Marginal inference task**. As presented in Figure 6a, all variations of multi-module GNNs exhibit a similar dual band-like pattern, resulting from the composition of training data including both $\mathcal{G}_2$ and $\mathcal{G}_4$. These patterns, however, are more prominent than that of the single-module baseline (Figure 6a; the outcome of

Table 2: OOD performance on 27 datasets. All numerical results are averages across 3 random runs. Numbers in **bold** represent the best results. Additional results and descriptions are in Appendix A.6 and A.7.

| Dataset | CLRS ↑ | GOOD-CMNIST ↑ | | | GOOD-Motif ↑ | | | | | GOOD-ZINC ↓ | | | | |
|---|---|---|---|---|---|---|---|---|---|---|---|---|---|---|
| Metric | Accuracy | Accuracy | | | Accuracy | | | | | MAE | | | | |
| Domain | - | color | | | base | | | size | | scaffold | | | size | |
| Dist. shift | - | covariate | concept | no shift | covariate | concept | no shift | covariate | concept | covariate | concept | no shift | covariate | concept |
| Aggregation | max | max | sum | sum | sum | sum | sum | sum | sum | max | max | max | max | max |
| MPNN | 67.22 | 35.34 | **77.57** | 92.93 | 63.68 | 86.94 | 92.53 | 57.32 | 72.87 | 0.3307 | 0.2186 | 0.1731 | **0.6297** | 0.2488 |
| Sigmoid-G | 66.68 | 34.71 | 66.99 | 91.90 | 58.43 | 87.17 | **92.57** | 53.29 | 72.19 | 0.3163 | 0.2182 | **0.1684** | 0.6421 | **0.2452** |
| Binary-G | **70.16** | **49.22** | 74.23 | 92.56 | 59.22 | **87.77** | 92.52 | 52.24 | 71.91 | **0.3146** | **0.2178** | 0.1719 | 0.6467 | 0.2506 |
| Meta-L | 67.38 | 43.61 | 77.53 | 92.61 | **76.48** | 87.17 | 92.45 | **58.69** | **73.38** | 0.3288 | 0.2285 | 0.1791 | 0.6696 | 0.2603 |

| Dataset | GOOD-SST2 ↑ | | | GOOD-PCBA ↑ | | | | | GOOD-HIV ↑ | | | | | # of tasks with the best results | |
|---|---|---|---|---|---|---|---|---|---|---|---|---|---|---|---|
| Metric | Accuracy | | | Average Precision | | | | | ROC-AUC | | | | | | |
| Domain | length | | | scaffold | | | size | | scaffold | | | size | | | |
| Dist. shift | covariate | concept | no shift | covariate | concept | no shift | covariate | concept | covariate | concept | no shift | covariate | concept | | |
| Aggregation | sum | max | max | sum | sum | sum | max | sum | sum | sum | max | sum | max | | |
| MPNN | 80.74 | **68.27** | 90.03 | 11.71 | 17.71 | 26.51 | **11.74** | 12.04 | **68.93** | 63.09 | **79.70** | 57.53 | 70.90 | 8/27 | 8/27 |
| Sigmoid-G | 80.45 | 65.79 | 90.16 | 11.70 | 17.89 | **27.02** | 10.95 | 11.56 | 65.20 | 63.54 | 78.73 | **59.05** | **71.32** | 6/27 | |
| Binary-G | 80.83 | 66.69 | 90.14 | 11.84 | **17.98** | 25.56 | 10.87 | 11.98 | 64.83 | 63.82 | 79.15 | 57.03 | 70.00 | 6/27 | **19/27** |
| Meta-L | **81.26** | 67.60 | **90.22** | **12.13** | 16.99 | 25.81 | 10.80 | 11.01 | 67.79 | **65.91** | 79.24 | 55.50 | 69.47 | 7/27 | |

the baseline is identical to Figure 5a (middle), but the color range has been adjusted to match that of the multi-module GNNs), indicating that the multi-module framework is more effective in predicting marginals in comparison to the baseline. To quantify this consistency across multi-module strategies, we compute the average KL divergence loss by projecting the heat map onto the first principal axis, and plot the population level test loss with a spline kernel smoothing (Figure 6b). The loss curves are characterized by two local minima (valleys) that correspond to the well-predicted test graphs, but the improved performance by the proposed methods is further supported by the lower KL divergence at each valley of the loss landscapes (Figure 6b) and higher peaks at each mode of the degree histograms (Figure 6c), when compared to the baseline GNN. We note that the superiority is particularly noticeable in the second valley and mode where the single-module GNN struggles to generalize (Table 1).

**Graph theory multi-task**. Up until now, all quantitative observations have been derived from the task of estimating marginal probabilities. A subsequent question that arises is whether these findings can be generalized to other tasks. Here, we extend our analysis to the multi-task graph theory problem. The first notable experimental result is that the vertical band-shaped pattern observed in our previous analysis is still present in this multi-task scenario (Figure 6d). This supports our hypothesis that the average node degree in each mode of degree distribution should be considered as a crucial factor when assessing the generalizability of GNNs across different graph benchmark tasks. Secondly, we observe a significant gap in generalization performance between single and multi-module GNNs (Figure 6d-6f), which is in stark contrast to the results from the marginal inference task. The higher effectiveness of multi-module GNNs over the baseline (Table 1) suggests that using a multi-module framework could be more beneficial when the task is not computationally complex (i.e. NP-hard) and ground-truth targets can be generated reliably.

Finally, our analysis also reveals that there is a marked difference in performance between GNNs that use a gating mechanism and one that employs meta-learning. Our experimental findings presented in Table 1 suggest that the meta-learning approach could be more effective in dealing with the multimodal degree scenario. This raises the question of what makes the meta-learning approach superior even though all strategies use the same number of update modules, and the gating mechanism requires an additional processor for the gating variable. One potential explanation for this improved performance is that the meta-learning approach endows GNNs with multiple learning processes of different phases. Furthermore, our inner-loop optimization does not require changing the weights of the update modules, which sets it apart from most gradient-based meta-learning algorithms (Finn et al., 2017; Nichol et al., 2018). This distinction enables the GNN to combine multiple modules more flexibly and may contribute to the enhanced learning effect for combinatorial generalization.

### 5.4 Multi-module GNNs on real-world graphs

So far, our experimentation has centered on utilizing our random graph benchmark. However, we will now redirect our attention towards real-world graphs. In Section 3.3, we explored how real-world graphs are represented on a 2D scatter plot and found that the majority of real graph benchmarks display a relatively low degree and are characterized by a single elongated cluster. To gain a better understanding of the distribution shifts among the training, validation, and testing graphs, we plot their node degree distributions from each real benchmark dataset and confirm that there is no specific shift in the average degree, except for the CLRS benchmark (Figure A4). This indicates that real-world graphs do not often conform to our multi-mode degree distribution hypothesis, making it difficult to confidently assert that our proposed framework will work well in real-world situations. Despite this limitation, we evaluate our multi-module GNNs using the CLRS and GOOD benchmarks.

In CLRS, we use random scalars in conjunction with deterministic positional indices as node features to facilitate training and expedite convergence (Veličković et al., 2022; Mahdavi et al., 2023), while we employ original node features and targets from the GOOD benchmark without any modifications. We use two different aggregators (max or sum) across all tasks and report the best outcome achieved among these two aggregations. Every experiment is repeated three times and averaged over three random seeds. Table 2 illustrates the average performance of the baseline and multi-module GNNs across a variety of real-world graph tasks. Despite some variations in performance across different prediction tasks, our proposed framework surpasses the baseline in 11 additional tasks (with 19 wins compared to 8), demonstrating the practical applicability of our proposed methods and findings. Moreover, we also include two well-established OOD generalization methods, namely invariant risk minimization (IRM) (Arjovsky et al., 2019) and variance risk extrapolation (VREx) (Krueger et al., 2021), in Table A6. In this experiment, the baseline, multi-module GNNs, and existing OOD generalization methods perform the best on 6, 13, and 8 tasks, respectively. Although these additional experiments yield mixed results, they continue to provide encouragement by demonstrating the predictive power of multi-module GNNs in specific tasks. Additional details on the tasks, results and analysis can be found in Appendices A.5-A.8.

## 6 Related Work

**OOD generalization on graphs**. The field of OOD generalization in graph domains has only recently gained attention from researchers. To address Graph OOD problems, four different approaches have been explored. The first involves using graph augmentation methods to increase the quantity and diversity of training samples (Zhao et al., 2021; Park et al., 2021; Wu et al., 2022a; Feng et al., 2020; Kong et al., 2022; Liu et al., 2022), including various forms of Mixup (Zhang et al., 2018; Verma et al., 2019) specifically tailored to the graph domain. These generate new instances by feature or graph structure interpolation (Verma et al., 2021; Wang et al., 2021; 2020; Han et al., 2022). The second approach aims to create disentangled representations with properties that enhance OOD generalization, such as GNNs with neighborhood routing mechanisms (Ma et al., 2019; Liu et al., 2020) or learnable edge masks (Fan et al., 2022). Self-supervised contrastive learning (Li et al., 2021; 2022a) is also used to eliminate the need for computationally expensive graph reconstruction steps. The third strategy considers the structural causal model in model design to capture the causal relations between input graph data and stable label distributions under distribution shifts (Sui et al., 2022; Chen et al., 2022; Bevilacqua et al., 2021; Zhou et al., 2022). A few initial studies have examined OOD generalization using graphon models (Bevilacqua et al., 2021; Maskey et al., 2022; Zhou et al., 2022), but they rely on the implicit assumption that subgraphs created by a subset of a graph's nodes will have unchanged labels, which may not always hold true. Finally, the invariant learning approach aims to leverage invariant relationships between features and labels across different distributions while ignoring variant spurious correlations (Li et al., 2022b; Wu et al., 2022c; Miao et al., 2022; Wu et al., 2022b; Zhang et al., 2022). This approach treats the cause of distribution shifts as an unknown environmental variable. Within our study, we view the mode of degree distribution as an environmental variable that can pose challenges for generalization, while graph size and other graph measures, except for the average degree, are regarded as invariant features. Consequently, our objective was to enhance generalization specifically with regard to these invariant features, rather than striving for uniform performance across multiple environments.

**Update module in GNNs**. The update function in GNNs has received relatively less attention from researchers than the aggregate function (Kipf & Welling, 2016; Zaheer et al., 2017; Hamilton et al., 2017a; Veličković et al., 2018; Murphy et al., 2018; Corso et al., 2020), although it plays an equally important role in defining the inductive bias of GNNs. Previous work on improving the expressive power of the update function can be distinguished by how the past node representations are used when updating themselves. Hamilton et al. (2017b) concatenated the output of the base update function with the past node representation, whereas Pham et al. (2017) linearly interpolated them to preserve node information from previous rounds of message passing during the update step. In a parallel line of work, the gated updates through recurrent neural network (RNN) based architectures have also been proposed to facilitate the learning of deep GNN models (Li et al., 2015; Selsam et al., 2019). These studies generally share the parameters of the update function across nodes and message-passing layers. However, such a single type of update module could result in restricted expressivity, and the effect of using multiple update modules are not explored in those works.

In a related research direction, Luan et al. (2020; 2022) suggested employing multiple channel filter banks to enrich node representations. This concept aligns with the idea of utilizing multiple message functions within the MPNN framework to generate different types of incoming messages. In this case, the node representation is obtained by combining multiple aggregated messages through a weighted sum, followed by a simple activation function. Both Luan et al. (2020; 2022) and our proposed method employ multiple nonlinear transformations; however, the dissimilarity lies in whether the nonlinear transformation is applied to the message function or the update function. Furthermore, Luan et al. (2020; 2022) specifically focuses on improving node classification tasks on heterophilic graphs using a sigmoid gating strategy, whereas our approach avoids limiting itself to particular task types and explores various ways of leveraging multiple modules. While previous approaches have primarily concentrated on methodologies within a single GNN layer, Chien et al. (2021); Eliasof et al. (2022) have shifted their focus towards enhancing the quality of the final node representations by adaptively leveraging the outputs of each propagation step, rather than solely relying on the final layer output.

## 7 Conclusion

In this work, we investigated the influence of the graph structure on the generalization performance of graph neural networks. Specifically, we used random graph generators to examine the impact of the different graph properties, such as degree, clustering coefficient, and path length, which led to useful findings such that OOD generalization ability of a GNN is highly correlated with the average degree of the graphs. To address the challenges posed by multi-mode degree distribution, we proposed a novel GNN architecture with multiple update modules and a meta-learning method to select the optimal aggregation strategy to improve OOD generalization, whose effectiveness have been validated on synthetic and real-world graph datasets.

**Acknowledgments**

This work was supported in part by the National Research Foundation of Korea (NRF) under Grant NRF-2018R1C1B5086404 and Grant NRF-2021R1F1A1045390; in part by the Brain Convergence Research Program of the National Research Foundation (NRF) through the Korean Government Ministry of Science and ICT (MSIT) under Grant NRF-2021M3E5D2A01023887; in part by Samsung Electronics Co., Ltd.

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

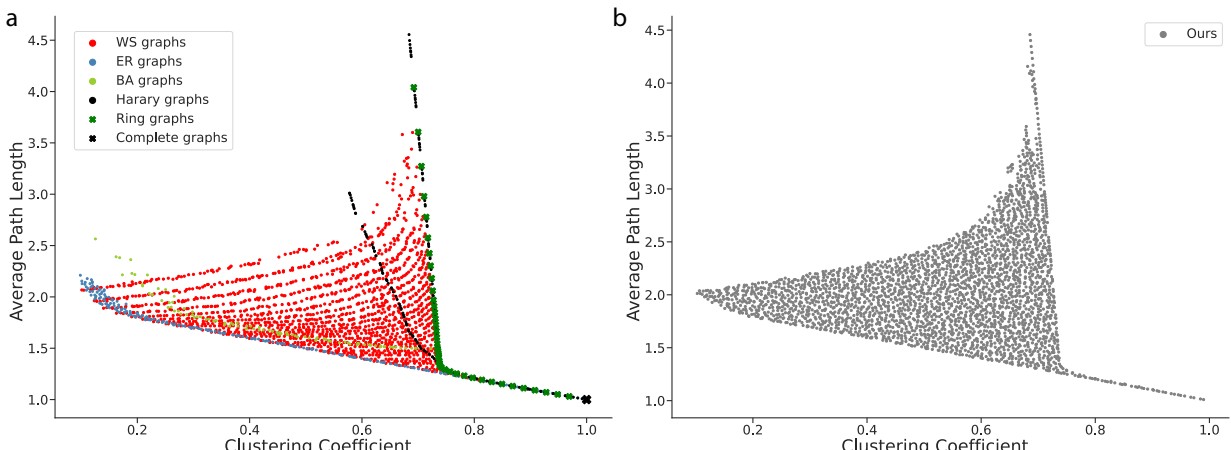

Figure A1: **Coverage of random graphs.** **(a)** Graphs sampled from four distinct random graph generators, along with ring and complete graphs having a size of $|\mathcal{V}| = 100$. **(b)** Random graphs generated through our method, which covers a more extensive area of the graph space.

# A    Appendix

## A.1    Details for generating random graphs

Here we provide more details for how we generated random graphs in Section 3.1. The relaxed Watts-Strogatz (WS) graphs are parameterized by: (1) number of nodes $n$, (2) average node degree $k$ (real number), and (3) edge rewiring probability (randomness) $p$. We search over:

- degree $k \in$ `np.linspace(2,n-2,300)`

- randomness $p \in$ `np.linspace(0,1,300)**2`

- 40 random seeds

To create more evenly distributed samples, we square $p$ and generate $300{\times}300{\times}40 = 3{,}600{,}000$ random graphs in total. This helps to mitigate the significant irregularities in graph properties that occur when $p$ is small. Then we create 2-dimensional square bins on the two principal axes of a 2D scatter plot (Figure 2b) and sample one graph from each bin, if any. These sampled graphs are then used for testing as described below:

- $|V| = $16:  3.6M to 1,282 graphs  (# bins $= 250{\times}250$)

- $|V| = $36:  3.6M to 1,443 graphs  (# bins $= 80{\times}80$)

- $|V| = $100:  3.6M to 1,584 graphs  (# bins $= 50{\times}50$)

## A.2    Model and optimization hyperparameters

Our baseline GNN processor utilizes the message-passing neural network (MPNN) and integrates the graph attention mechanism when suitable for enhancing generalization performance. While there are numerous types of GNN layers available, we chose MPNN due to its adaptable architecture that consists of different functional modules. Thus, the pertinent message, update, and aggregate functions are determined by one of the fundamental neural network modules, depending on the requirement that the processor is capable of reasonably solving the given task. Apart from these components, we used the same model (Table A1) and optimization (Table A2) hyperparameters as in prior research.

Table A1: Model hyperparameters for synthetic and real-world graph benchmarks.

| | processor | message function | update function | gating function | aggregation $\oplus$ | embedding dimension | # layers |
|---|---|---|---|---|---|---|---|
| Marginal inference | GAT | MLP | GRU | MLP | sum | 64 | 1 |
| Graph theory multitask | MPNN | Linear | GRU | MLP | max | 16 | 2 |
| CLRS benchmark | MPNN | MLP | GRU | MLP | max | 128 | 1 |
| GOOD benchmark | MPNN | MLP | MLP | MLP | max/sum | 300 | varied on each task |

Table A2: Optimization hyperparameters for synthetic and real-world graph benchmarks.

| | max epoch | learning rate | batch size | optimizer | weight decay | loss function | # propagation | # samples in meta-testing |
|---|---|---|---|---|---|---|---|---|
| Marginal inference | 1000 | 0.001 | 32 | Adam | 0.0 | KL Divergence | 10 | 5 for $\mathcal{G}_i$ |
| Graph theory multitask | 5000 | 0.003 | 256 | Adam | $1e^{-6}$ | MSE | 1 | 5 for $\mathcal{G}_i$ |
| CLRS benchmark | 640 | 0.0001 | 32 | Adam | 0.0 | MSE(scalar), Cross-entropy(categorical) | 32 | 5 for $\mathcal{G}_i$ |
| GOOD benchmark | 200, 500(CMNIST) | 0.001 | varied on each dataset | Adam | 0.0 | MAE(scalar), Cross-entropy(categorical) | varied on each dataset | 5 for $\mathcal{G}_i$ |

Table A3: Summary of CLRS dataset configurations.

| Dataset | # graph nodes train/val | test | # graphs train | val/test | Graph generator |
|---|---|---|---|---|---|
| CLRS | 16 | 64 | 1000 | 32 | Erdős-Rényi graphs with fixed $p$ |

Table A4: Number of graphs in training, ID validation, ID testing, and OOD testing sets for 6 graph-level tasks in GOOD benchmark.

| Dataset | Shift | Train | ID validation | ID test | OOD test | Train | ID validation | ID test | OOD test |
|---|---|---|---|---|---|---|---|---|---|
| | | | | Scaffold | | | | Size | |
| | covariate | 24682 | 4112 | - | 4108 | 26169 | 4112 | - | 3961 |
| GOOD-HIV | concept | 15209 | 3258 | - | 10037 | 14454 | 3096 | - | 10525 |
| | no shift | 24676 | 8225 | 8226 | - | 25676 | 8225 | 8226 | - |
| Dataset | Shift | Train | ID validation | ID test | OOD test | Train | ID validation | ID test | OOD test |
| | | | | Scaffold | | | | Size | |
| | covariate | 262764 | 43792 | - | 43562 | 269990 | 43792 | - | 31925 |
| GOOD-PCBA | concept | 159158 | 34105 | - | 119821 | 150121 | 32168 | - | 115205 |
| | no shift | 262757 | 87586 | 87586 | - | 262757 | 87586 | 87586 | - |
| Dataset | Shift | Train | ID validation | ID test | OOD test | Train | ID validation | ID test | OOD test |
| | | | | Scaffold | | | | Size | |
| | covariate | 149674 | 24945 | - | 24946 | 161893 | 24945 | - | 17402 |
| GOOD-ZINC | concept | 101867 | 21828 | - | 60393 | 89418 | 19161 | - | 70306 |
| | no shift | 149673 | 49891 | 49891 | - | 149673 | 49891 | 49891 | - |
| Dataset | Shift | Train | ID validation | ID test | OOD test | Train | ID validation | ID test | OOD test |
| | | | | Length | | | | | |
| | covariate | 24744 | 5301 | - | 17490 | | | | |
| GOOD-SST2 | concept | 27270 | 5843 | - | 15944 | | | | |
| | no shift | 42025 | 14008 | 14009 | - | | | | |
| Dataset | Shift | Train | ID validation | ID test | OOD test | Train | ID validation | ID test | OOD test |
| | | | | Color | | | | | |
| | covariate | 42000 | 7000 | - | 7000 | | | | |
| GOOD-CMNIST | concept | 29400 | 6300 | - | 14000 | | | | |
| | no shift | 42000 | 14000 | 14000 | - | | | | |
| Dataset | Shift | Train | ID validation | ID test | OOD test | Train | ID validation | ID test | OOD test |
| | | | | Base | | | | Size | |
| | covariate | 18000 | 3000 | - | 3000 | 18000 | 3000 | - | 3000 |
| GOOD-Motif | concept | 12600 | 2700 | - | 6000 | 12600 | 2700 | - | 6000 |
| | no shift | 18000 | 6000 | 6000 | - | 18000 | 6000 | 6000 | - |

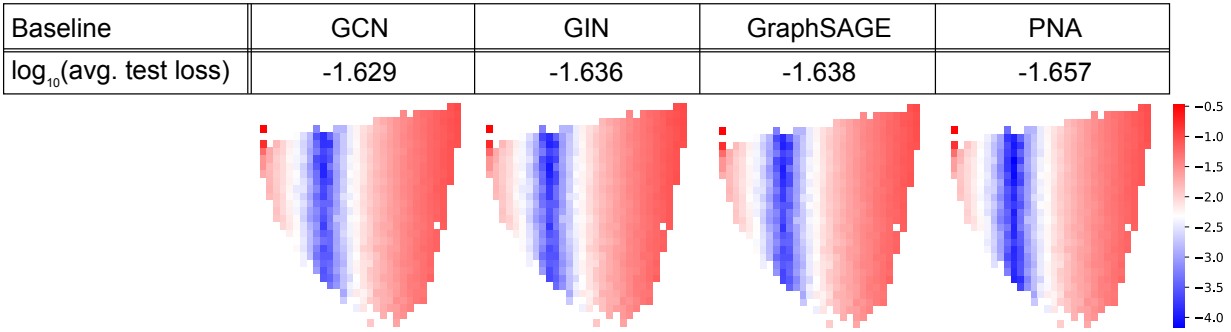

| Baseline | GCN | GIN | GraphSAGE | PNA |
|---|---|---|---|---|
| $\log_{10}$(avg. test loss) | -1.629 | -1.636 | -1.638 | -1.657 |

Figure A2: Generalization pattern in graph theory multitask observed from other GNN baseline models, when trained on $\mathcal{G}_2$.

### A.3  Effect of GNN processors on generalization patterns

To explore how other variants of GNNs generalize, we conducted an additional graph theory multitask experiment by training them on $\mathcal{G}_2$. The results depicted in Figure A2 indicate that GNNs' ability to generalize is consistently correlated with the average degree of the graphs, regardless of whether GCN, GIN, GraphSAGE, or PNA is used:

- GCN (Kipf & Welling, 2016):

$$\mathbf{h}'_v \quad = \quad \max_{u \in \mathcal{N}(v) \cup \{u\}} \frac{1}{\sqrt{|\mathcal{N}(v)||\mathcal{N}(u)|}} \mathbf{h}_u^{(l)} W^{(l)} \tag{7}$$

- GIN (Xu et al., 2019):

$$\mathbf{h}'_v \quad = \quad \sigma \left( (1 + \epsilon) \cdot \mathbf{h}_v^{(l)} + \max_{u \in \mathcal{N}(v)} \mathbf{h}_u^{(l)} \right) \tag{8}$$

- GraphSAGE (Hamilton et al., 2017a):

$$\mathbf{h}'_v \quad = \quad \mathbf{W}_1 \cdot \mathbf{h}_v^{(l)} + \mathbf{W}_2 \cdot \max_{u \in \mathcal{N}(v)} \mathbf{h}_u^{(l)} \tag{9}$$

- PNA (Corso et al., 2020):

$$\mathbf{h}'_v \quad = \quad \bigoplus_{u \in \mathcal{N}(v)} \mathbf{MLP} \left( \mathbf{h}_u^{(l)}, \mathbf{h}_v^{(l)} \right) \tag{10}$$

$$\text{where} \bigoplus \quad = \quad \underbrace{\begin{bmatrix} 1 \\ S(\mathbf{D}, \alpha = 1) \\ S(\mathbf{D}, \alpha = -1) \end{bmatrix}}_{\text{scalers}} \otimes \underbrace{\begin{bmatrix} \mu \\ \sigma \\ \max \\ \min \end{bmatrix}}_{\text{aggregators}} \tag{11}$$

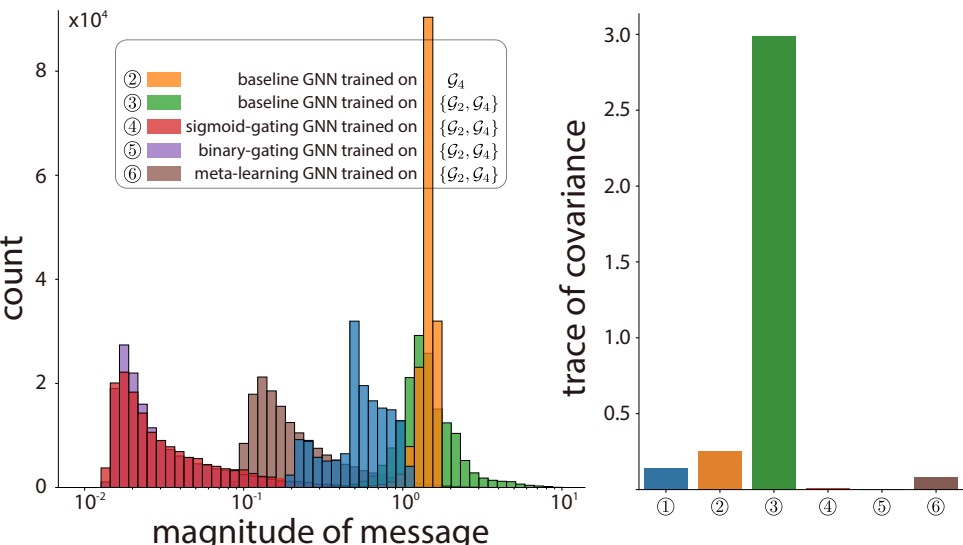

Figure A3: Message analysis from graph theory multitask.

## A.4 BOUNCEGRAD algorithm

When implementing the meta-learning method, it is essential to employ an algorithm that can effectively find approximately optimal update module assignment vectors. In this work, we implement a modified version of the BounceGrad algorithm (Alet et al., 2018) for the graph structured data, which takes the following inputs: a sample space $\mathbb{A}$ of update module assignments, a set of graphs $\mathbb{G}$, a set of network weights $\Theta$, a learning rate $\eta$, a temperature $T$, and a rate of change in temperature $\alpha \in (0, 1)$. As in the paper (Alet et al., 2018), we use simulated annealing, a local search algorithm involving stochasticity.

The algorithm begins with initializing the Simulated Annealing running accuracy rate $(\mathrm{SA}_r)$ and factor $(\mathrm{SA}_f)$. In the main loop, the temperature is updated by the ScheduleTemp procedure based on the current step, the rate of change in temperature $\alpha$, the $\mathrm{SA}_r$, and the $\mathrm{SA}_f$. Next, the BOUNCE procedure is called for updating the assignment $\mathcal{A}$. During this step, new assignments $\mathcal{A}'$ are proposed for graphs in the mini-batch, and their performance are compared to that of the current assignment using Accept function. The new assignment is accepted when it decreases the loss on the task or, with some probability based on current temperature, even if it does not. Finally, we call the Grad using the assignment to update the network parameters $\Theta$ via the backpropagation algorithm.

To explore the optimal structure of the modules and adapt to new graphs, we run 1000 iterations of meta-test phases for a small number of final training graphs, denoted as $\mathcal{G}^{train}_{meta-test}$. During the meta-testing phase, we rely on the Bounce procedure given a fixed $\Theta$. This approach enables us to find the optimal assignment of the update modules for few-shot meta-test graphs.

---

**Algorithm 1** BounceGrad

---

1: **procedure** BounceGrad($\mathbb{A}, \mathcal{G}_{train}^{i=1:N}, \mathcal{G}_{test}^{i=1:N}, \text{SA}_r, \text{SA}_f, \eta, T, \alpha$)
2:     Initialize $\text{SA}_r, \text{SA}_f$
3:     **for** $i \leftarrow 0$ to $\text{epoch}_{max}$ **do**
4:        $T \leftarrow \text{ScheduleTemp}(T, i/\text{epoch}_{max}, \alpha, \text{SA}_r, \text{SA}_f)$
5:        $\mathcal{A}_{i=1:N}, \text{SA}_r, \text{SA}_f \leftarrow \text{Bounce}(\mathcal{A}_{i=1:N}, \mathcal{G}_{train}^{i=1:N}, T, \mathbb{A}, \Theta, \text{SA}_r, \text{SA}_f)$
6:        $\Theta \leftarrow \text{Grad}(\Theta, \mathcal{A}_{i=1:N}, \mathcal{G}_{test}^{i=1:N}, \eta)$

---

**Algorithm 2** ScheduleTemp

---

1: **procedure** ScheduleTemp($T, step, \alpha, \text{SA}_r, \text{SA}_f$)
2:     $\text{ACC} \leftarrow \exp\left(-5 * step\right)$
3:     **if** $\text{SA}_r/\text{SA}_f < \text{ACC}$ **then**
4:        $T \leftarrow T * \alpha$
5:     **else**
6:        $T \leftarrow T/\alpha$
    **return** $T$

---

**Algorithm 3** Bounce

---

1: **procedure** Bounce($\mathcal{A}_{j=1:N}, \mathcal{G}_{train}^{j=1:N}, T, \mathbb{A}, \Theta, \text{SA}_r, \text{SA}_f$)
2:     **for** $j \leftarrow 1$ to $N$ **do**
3:        $\mathcal{A}'_j \leftarrow \text{Propose}_{\mathbb{A}}(\mathcal{A}_j, \Theta)$
4:        $\ell' \leftarrow \ell(\mathcal{G}_{train}^j, \mathcal{A}'_j, \Theta)$
5:        $\ell \leftarrow \ell(\mathcal{G}_{train}^j, \mathcal{A}_j, \Theta)$
6:        $f \leftarrow \min(10^{-2}, \text{SA}_r/\text{SA}_f)$
7:        **if** $\text{Accept}(\ell', \ell, T)$ **then**
8:           $\mathcal{A}_j \leftarrow \mathcal{A}'_j$
9:           **if** $\ell' >= \ell$ **then**
10:             $\text{SA}_f \leftarrow (1 - f) * \text{SA}_f + f$
11:             $\text{SA}_r \leftarrow (1 - f) * \text{SA}_r + f$
12:        **else**
13:           $\text{SA}_f \leftarrow (1 - f) * \text{SA}_f + f$
14:           $\text{SA}_r \leftarrow (1 - f) * \text{SA}_r$
15:     **return** $\mathcal{A}_{j=1:N}, \text{SA}_r, \text{SA}_f$

---

**Algorithm 4** Accept

---

1: **procedure** Accept($\ell, \ell', T$)
2:     **return** $\ell' < \ell$ or $\text{rand}(0, 1) < \exp\{(\ell - \ell')/T\}$

---

---

**Algorithm 5** Grad

---

1: **procedure** $\text{GRAD}(\Theta, \mathcal{A}_{i=1:N}, \mathcal{G}_{test}^{i=1:N}, \eta)$
2:      $\Delta = 0$
3:      **for** $j \leftarrow 1$ to $N$ **do**
4:          $(x, y) = \text{rand\_elt}\,(\mathcal{G}_{test}^j)$
5:          $\Delta = \Delta + \nabla_\Theta \mathcal{L}(\mathcal{A}_{j\Theta}(x), y)$
6:      $\Theta = \Theta - \eta \Delta$
7: **return** $\Theta$

---

```python
def accept(loss_prev, loss_new, t):
    """Decide whether to accept the new proposal based on the loss."""
    delta = loss_new - loss_prev
    if loss_new < loss_prev or np.random.rand() < np.exp(-delta / t):
        return True
    else:
        return False

def schedule_temp(s_r, s_f, t):
    """Update temperature adaptively using a geometric progression."""
    acc = np.exp(-5 * epoch / max_epoch)
    if s_r / s_f < acc:
        t *= alpha
    else:
        t /= alpha
    return t

def update_parameters(s_r, s_f, f, acceptance):
    """Update SA parameters based on the success rate."""
    s_r = (1.0 - f) * s_r + f if acceptance == True else (1.0 - f) * s_r
    s_f = (1.0 - f) * s_f + f
    return s_r, s_f

def bounce(assignment_prev, train_data, T, s_r, s_f, f):
    """Explore more optimal update module structures using simulated annealing."""
    x, y = train_data.x, train_data.y
    assignment_new = propose(assignment_prev) # Propose a new update module structure.
    out1, out2 = model(x, a_prev), model(x, a_new) # Evaluate the current and proposed
                                                   structures.
    if accept(loss(out1, y), loss(out2, y), T):
        if loss(out2, y) >= loss(out1, y):
            s_r, s_f = update_parameters(s_r, s_f, f, True)
    else:
        assignment_new = assignment_prev
        s_r, s_f = update_parameters(s_r, s_f, f, False)
    return assignment_new, s_r, s_f

def grad(assignment, test_data):
    """Update model weights based on the test graphs."""
    x, y = test_data.x, test_data.y
    out = model(x, assignment)
    loss(out, y).backward()

# Initialize update module assignment.
assignment = np.random.choice([0, 1], size=(n, num_nodes))
# Loop over minibatches
for train_data, test_data in zip(train_loader, test_loader):
    T = schedule_temp(s_r, s_f, T)  # Update SA temperature.
    # Explore alternative module structures (bounce) and assess their effectiveness (grad).
    assignment, s_r, s_f = bounce(assignment, train_data, T, s_r, s_f, min(0.01, s_r / s_f))
    grad(assignment, test_data)
```

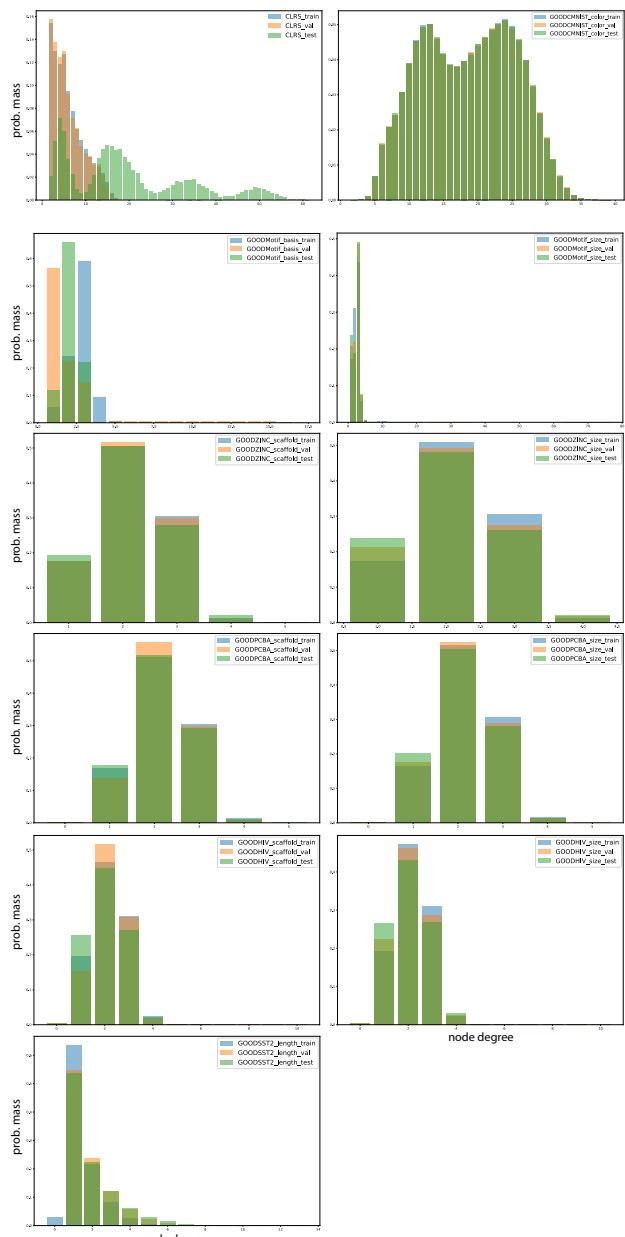

Figure A4: Normalized node degree histograms from CLRS and GOOD benchmarks.

## A.5    Details for real-world graph benchmark

The configurations for the CLRS and GOOD benchmark datasets are presented in Tables A3 and A4, respectively. Additionally, the average node degree distributions are compared among the training, validation, and testing graphs for each real-world graph benchmark (Figure A4). It is worth noting that the structural properties of most benchmark graphs are not clearly distinguishable, as shown in Figure 2d.

## A.6    Complete numerical results for CLRS benchmark

Prior research (Mahdavi et al., 2023) has been referred to create our experimental setup, which suggests that disabling hint trajectories can lead to improved performance. When hints are disabled, two algorithms from the same category exhibit identical behavior. As a result, the number of tasks has been reduced from 30 to

24 by choosing one from each algorithmic category. To expedite convergence and facilitate training, we use random scalars in conjunction with deterministic positional indices as node features. Our experiments are repeated three times, and the results are averaged over three random seeds. Table 2 presents the average test scores for all 24 algorithmic tasks, while Table A5 provides the mean and standard deviation for individual tasks.

Table A5: Complete test accuracy (%) of MPNN and three multi-module GNN strategies on CLRS benchmark over 24 algorithmic reasoning tasks.

| Algorithms | MPNN | Sigmoid-G | Binary-G | Meta-L |
|---|---|---|---|---|
| Activity Selector | 69.11 ± 14.67 | 65.68 ± 13.56 | **87.74** ± 4.90 | 71.02 ± 1.69 |
| Articulation Points | 95.82 ± 0.58 | 96.41 ± 0.79 | **97.01** ± 0.38 | 96.75 ± 0.48 |
| Bellman-Ford | 98.60 ± 0.33 | **98.70** ± 0.39 | 98.36 ± 0.33 | 71.02 ± 0.00 |
| BFS | 99.97 ± 0.03 | 99.93 ± 0.03 | 99.98 ± 0.03 | **100.00** ± 0.00 |
| Binary Search | 92.46 ± 1.56 | 91.88 ± 1.12 | **93.88** ± 2.13 | 89.79 ± 2.13 |
| Bridges | 59.51 ± 11.75 | 56.67 ± 4.52 | **87.36** ± 6.02 | 75.41 ± 8.60 |
| DAG Shortest Paths | 99.22 ± 0.27 | 99.28 ± 0.20 | **99.54** ± 0.20 | 99.17 ± 0.08 |
| DFS | 25.46 ± 1.27 | 24.66 ± 2.83 | **27.65** ± 1.67 | 25.63 ± 2.75 |
| Find Max. Subarray | **28.74** ± 5.94 | 24.48 ± 1.82 | 27.05 ± 1.29 | 23.76 ± 2.54 |
| Floyd-Warshall | **31.28** ± 1.55 | 29.29 ± 1.02 | 31.18 ± 1.49 | 27.78 ± 0.63 |
| Graham Scan | 99.00 ± 0.07 | **99.04** ± 0.31 | 98.90 ± 0.12 | 98.80 ± 0.52 |
| LCS Length | 58.45 ± 1.55 | 58.92 ± 1.30 | **60.08** ± 2.29 | 58.53 ± 0.93 |
| Matrix Chain Order | 81.13 ± 3.99 | 77.10 ± 3.92 | **81.55** ± 5.61 | 79.74 ± 2.81 |
| Minimum | **99.64** ± 0.20 | 99.59 ± 0.11 | 99.40 ± 0.10 | 99.04 ± 0.30 |
| MST-Kruskal | 59.82 ± 1.13 | 61.65 ± 1.95 | **61.71** ± 3.03 | 59.49 ± 0.87 |
| MST-Prim | **83.28** ± 0.71 | 82.88 ± 1.37 | 82.88 ± 1.59 | 79.67 ± 1.54 |
| Naïve String Match | 10.69 ± 1.95 | 10.71 ± 4.04 | **21.70** ± 8.88 | 5.31 ± 1.40 |
| Optimal BST | **73.92** ± 2.10 | 73.20 ± 1.29 | 71.34 ± 5.18 | 70.14 ± 2.35 |
| Quickselect | 0.00 ± 0.00 | 2.51 ± 4.34 | **8.61** ± 8.21 | 0.00 ± 0.00 |
| Quicksort | 8.15 ± 2.74 | 8.38 ± 4.42 | 12.68 ± 2.45 | **19.06** ± 4.92 |
| Segments Intersect | **98.52** ± 0.10 | 98.47 ± 0.29 | **98.52** ± 0.16 | 98.18 ± 0.11 |
| SCC | 72.49 ± 0.60 | **73.81** ± 1.18 | 71.26 ± 1.30 | 72.79 ± 0.56 |
| Task Scheduling | **83.98** ± 0.82 | 83.33 ± 0.47 | 83.87 ± 0.40 | 83.50 ± 0.46 |
| Topological Sort | 83.92 ± 1.21 | 83.85 ± 0.65 | 81.65 ± 2.81 | **85.08** ± 1.66 |
| Overall average | 67.22 | 66.68 | **70.16** | 67.38 |

## A.7 Complete numerical results for Graph Out-Of-Distribution (GOOD) benchmark

The GOOD benchmark comprises several real-world graph datasets that exhibit unique domain shifts. It provides covariate and concept shifts by altering the input distributions and target distribution, respectively, between the training and testing data. Each dataset contains domain-specific shifts, such as graph size and scaffold in molecular graphs, that cover both types of distribution shifts. Some datasets have no shift, representing the original dataset. Our study focuses solely on graph prediction tasks, as the node-level tasks are based on a single graph (i.e., transductive setting), which is beyond the scope of our research aimed at investigating the ability to generalize to unseen graphs across multiple graphs. The tables A6-A16 below provide information on the mean and standard deviation of OOD generalization performance for each domain shift from the MPNN baseline, multi-module GNNs, and existing OOD methods (IRM and VREx). For more detailed information, please refer to Gui et al. (2022).

Table A6: OOD performance on 27 datasets. All numerical results are averages across 3 random runs. Numbers in **bold** represent the best results.

| Dataset | CLRS ↑ | GOOD-CMNIST ↑ | | | GOOD-Motif ↑ | | | | | GOOD-ZINC ↓ | | | | |
| --- | --- | --- | --- | --- | --- | --- | --- | --- | --- | --- | --- | --- | --- | --- |
| Metric | Accuracy | Accuracy | | | Accuracy | | | | | MAE | | | | |
| Domain | - | color | | | base | | | size | | scaffold | | | size | |
| Dist. shift | - | covariate | concept | no shift | covariate | concept | no shift | covariate | concept | covariate | concept | no shift | covariate | concept |
| Aggregation | max | max | sum | sum | sum | sum | sum | sum | sum | max | max | max | max | max |
| MPNN | 67.22 | 35.34 | **77.57** | **92.93** | 63.68 | 86.94 | 92.53 | 57.32 | 72.87 | 0.3307 | 0.2186 | 0.1731 | **0.6297** | 0.2488 |
| Sigmoid-G | 66.68 | 34.71 | 66.99 | 91.90 | 58.43 | 87.17 | **92.57** | 53.29 | 72.19 | 0.3163 | 0.2182 | **0.1684** | 0.6421 | 0.2452 |
| Binary-G | **70.16** | **49.22** | 74.23 | 92.56 | 59.22 | **87.77** | 92.52 | 52.24 | 71.91 | 0.3146 | **0.2178** | 0.1719 | 0.6467 | 0.2506 |
| Meta-L | 67.38 | 43.61 | 77.53 | 92.61 | **76.48** | 87.17 | 92.45 | 58.69 | 73.38 | 0.3288 | 0.2285 | 0.1791 | 0.6696 | 0.2603 |
| IRM | - | 41.97 | 76.45 | 92.61 | 68.07 | 87.33 | 92.48 | 56.14 | 73.61 | **0.3088** | 0.2207 | 0.1717 | 0.6338 | 0.2480 |
| VREx | - | 45.58 | 73.83 | 92.80 | 73.73 | 87.69 | 92.53 | **59.72** | **74.07** | 0.3158 | 0.2182 | 0.1723 | 0.6300 | **0.2429** |

| Dataset | GOOD-SST2 ↑ | | | GOOD-PCBA ↑ | | | | | GOOD-HIV ↑ | | | | | # of tasks with the best results | |
| --- | --- | --- | --- | --- | --- | --- | --- | --- | --- | --- | --- | --- | --- | --- | --- |
| Metric | Accuracy | | | Average Precision | | | | | ROC-AUC | | | | | | |
| Domain | length | | | scaffold | | | size | | scaffold | | | size | | | |
| Dist. shift | covariate | concept | no shift | covariate | concept | no shift | covariate | concept | covariate | concept | no shift | covariate | concept | | |
| Aggregation | sum | max | max | sum | sum | sum | max | sum | max | sum | max | sum | max | | |
| MPNN | 80.74 | **68.27** | 90.03 | 11.71 | 17.71 | 26.51 | 11.74 | **12.04** | **68.93** | 63.09 | 79.70 | 57.53 | 70.90 | 6/27 | 6/27 |
| Sigmoid-G | 80.45 | 65.79 | 90.16 | 11.70 | 17.89 | **27.02** | 10.95 | 11.56 | 65.20 | 63.54 | 78.73 | 59.05 | **71.32** | 4/27 | |
| Binary-G | 80.83 | 66.69 | 90.14 | 11.84 | **17.98** | 25.56 | 10.87 | 11.98 | 64.83 | 63.82 | 79.15 | 57.03 | 70.00 | 5/27 | **13/27** |
| Meta-L | **81.26** | 67.60 | **90.22** | 12.13 | 16.99 | 25.81 | 10.80 | 11.01 | 67.79 | **65.91** | 79.24 | 55.50 | 69.47 | 4/27 | |
| IRM | 79.79 | 65.82 | 90.09 | 11.92 | 17.41 | 26.31 | 11.64 | 11.65 | 65.07 | 62.15 | 78.42 | **62.22** | 70.27 | 2/26 | |
| VREx | 78.98 | 67.30 | 90.14 | **12.24** | 16.25 | 26.91 | **11.75** | 11.95 | 68.02 | 63.23 | **79.97** | 56.73 | 70.20 | 6/26 | 8/26 |

Table A7: Performance on GOOD-CMNIST with color domain

| **Accuracy** | Aggregation | MPNN | Sigmoid-G | Binary-G | Meta-L | IRM | VREx |
| --- | --- | --- | --- | --- | --- | --- | --- |
| covariate | max | $35.34 \pm 15.01$ | $34.71 \pm 11.14$ | **49.22** $\pm 3.85$ | $43.61 \pm 29.35$ | $41.97 \pm 18.06$ | $45.58 \pm 12.32$ |
| concept | sum | **77.57** $\pm 0.72$ | $66.99 \pm 6.78$ | $74.23 \pm 0.74$ | $77.53 \pm 0.61$ | $76.45 \pm 1.38$ | $73.83 \pm 4.61$ |
| no shift | sum | **92.93** $\pm 0.11$ | $91.90 \pm 0.83$ | $92.56 \pm 0.09$ | $92.61 \pm 0.15$ | $92.61 \pm 0.46$ | $92.80 \pm 0.17$ |

Table A8: Performance on GOOD-Motif with base domain

| Accuracy | Aggregation | MPNN | Sigmoid-G | Binary-G | Meta-L | IRM | VREx |
|---|---|---|---|---|---|---|---|
| covariate | sum | $63.68 \pm 2.50$ | $58.43 \pm 0.76$ | $59.22 \pm 1.75$ | $\mathbf{76.48} \pm 3.61$ | $68.07 \pm 2.48$ | $73.73 \pm 6.48$ |
| concept | sum | $86.94 \pm 0.69$ | $87.17 \pm 0.62$ | $\mathbf{87.77} \pm 1.14$ | $87.17 \pm 1.62$ | $87.33 \pm 0.25$ | $87.69 \pm 1.04$ |

Table A9: Performance on GOOD-Motif with size domain

| Accuracy | Aggregation | MPNN | Sigmoid-G | Binary-G | Meta-L | IRM | VREx |
|---|---|---|---|---|---|---|---|
| covariate | sum | $57.32 \pm 4.55$ | $53.29 \pm 4.06$ | $52.24 \pm 2.76$ | $58.69 \pm 12.37$ | $56.14 \pm 4.57$ | $\mathbf{59.72} \pm 3.78$ |
| concept | sum | $72.87 \pm 0.88$ | $72.19 \pm 2.45$ | $71.91 \pm 0.34$ | $73.38 \pm 2.43$ | $73.61 \pm 1.94$ | $\mathbf{74.07} \pm 2.04$ |
| no shift | sum | $92.53 \pm 0.01$ | $\mathbf{92.57} \pm 0.00$ | $92.52 \pm 0.08$ | $92.45 \pm 0.08$ | $92.48 \pm 0.07$ | $92.53 \pm 0.04$ |

Table A10: Performance on GOOD-ZINC with scaffold domain

| MAE | Aggregation | MPNN | Sigmoid-G | Binary-G | Meta-L | IRM | VREx |
|---|---|---|---|---|---|---|---|
| covariate | max | $0.3307 \pm 0.0076$ | $0.3163 \pm 0.0041$ | $0.3146 \pm 0.0043$ | $0.3288 \pm 0.0110$ | $\mathbf{0.3088} \pm 0.0001$ | $0.3158 \pm 0.0034$ |
| concept | max | $0.2186 \pm 0.0004$ | $0.2182 \pm 0.0036$ | $\mathbf{0.2178} \pm 0.0016$ | $0.2285 \pm 0.0041$ | $0.2207 \pm 0.0001$ | $0.2182 \pm 0.0001$ |
| no shift | max | $0.1731 \pm 0.0020$ | $\mathbf{0.1684} \pm 0.0027$ | $0.1719 \pm 0.0019$ | $0.1791 \pm 0.0010$ | $0.1717 \pm 0.0001$ | $0.1723 \pm 0.0011$ |

Table A11: Performance on GOOD-ZINC with size domain

| MAE | Aggregation | MPNN | Sigmoid-G | Binary-G | Meta-L | IRM | VREx |
|---|---|---|---|---|---|---|---|
| covariate | max | $\mathbf{0.6297} \pm 0.0134$ | $0.6421 \pm 0.0026$ | $0.6467 \pm 0.0045$ | $0.6696 \pm 0.0107$ | $0.6338 \pm 0.0092$ | $0.6300 \pm 0.0001$ |
| concept | max | $0.2488 \pm 0.0040$ | $0.2452 \pm 0.0027$ | $0.2506 \pm 0.0030$ | $0.2603 \pm 0.0079$ | $0.2480 \pm 0.0034$ | $\mathbf{0.2429} \pm 0.0001$ |

Table A12: Performance on GOOD-SST2 with length domain

| Accuracy | Aggregation | MPNN | Sigmoid-G | Binary-G | Meta-L | IRM | VREx |
|---|---|---|---|---|---|---|---|
| covariate | sum | $80.74 \pm 0.44$ | $80.45 \pm 0.87$ | $80.83 \pm 1.28$ | $\mathbf{81.26} \pm 1.19$ | $79.79 \pm 0.56$ | $78.98 \pm 2.99$ |
| concept | max | $\mathbf{68.27} \pm 1.76$ | $65.79 \pm 0.64$ | $66.69 \pm 0.95$ | $67.60 \pm 1.86$ | $65.82 \pm 0.00$ | $67.30 \pm 0.40$ |
| no shift | max | $90.03 \pm 0.18$ | $90.16 \pm 0.14$ | $90.14 \pm 0.01$ | $\mathbf{90.22} \pm 0.18$ | $90.09 \pm 0.00$ | $90.14 \pm 0.08$ |

Table A13: Performance on GOOD-PCBA with scaffold domain

| AP | Aggregation | MPNN | Sigmoid-G | Binary-G | Meta-L | IRM | VREx |
|---|---|---|---|---|---|---|---|
| covariate | sum | $11.71 \pm 0.09$ | $11.70 \pm 0.35$ | $11.84 \pm 0.26$ | $12.13 \pm 0.07$ | $11.92 \pm 0.22$ | $\mathbf{12.24} \pm 0.16$ |
| concept | sum | $17.71 \pm 0.55$ | $17.89 \pm 0.54$ | $\mathbf{17.98} \pm 1.03$ | $16.99 \pm 0.71$ | $17.41 \pm 0.63$ | $16.25 \pm 1.24$ |
| no shift | sum | $26.51 \pm 0.96$ | $\mathbf{27.02} \pm 0.50$ | $25.56 \pm 1.28$ | $25.81 \pm 0.51$ | $26.31 \pm 0.68$ | $26.91 \pm 0.87$ |

Table A14: Performance on GOOD-PCBA with size domain

| AP | Aggregation | MPNN | Sigmoid-G | Binary-G | Meta-L | IRM | VREx |
|---|---|---|---|---|---|---|---|
| covariate | max | $11.74 \pm 0.40$ | $10.95 \pm 0.30$ | $10.87 \pm 0.38$ | $10.80 \pm 0.31$ | $11.64 \pm 0.00$ | $\mathbf{11.75} \pm 0.64$ |
| concept | sum | $\mathbf{12.04} \pm 0.31$ | $11.56 \pm 0.63$ | $11.98 \pm 0.29$ | $11.01 \pm 0.34$ | $11.65 \pm 0.68$ | $11.95 \pm 0.30$ |

Table A15: Performance on GOOD-HIV with scaffold domain

| ROC-AUC | Aggregation | MPNN | Sigmoid-G | Binary-G | Meta-L | IRM | VREx |
|---|---|---|---|---|---|---|---|
| covariate | max | $\mathbf{68.93} \pm 2.15$ | $65.20 \pm 2.65$ | $64.83 \pm 2.17$ | $67.79 \pm 1.97$ | $65.07 \pm 1.60$ | $68.02 \pm 2.51$ |
| concept | sum | $63.09 \pm 3.11$ | $63.54 \pm 1.59$ | $63.82 \pm 1.31$ | $\mathbf{65.91} \pm 0.83$ | $62.15 \pm 1.73$ | $63.23 \pm 3.99$ |
| no shift | max | $79.70 \pm 0.46$ | $78.73 \pm 0.27$ | $79.15 \pm 0.31$ | $79.24 \pm 1.11$ | $78.42 \pm 0.74$ | $\mathbf{79.97} \pm 0.97$ |

Table A16: Performance on GOOD-HIV with size domain

| ROC-AUC | Aggregation | MPNN | Sigmoid-G | Binary-G | Meta-L | IRM | VREx |
|---|---|---|---|---|---|---|---|
| covariate | sum | $57.53 \pm 1.89$ | $59.05 \pm 0.92$ | $57.03 \pm 7.18$ | $55.50 \pm 0.33$ | $\mathbf{62.22} \pm 5.02$ | $56.73 \pm 1.47$ |
| concept | max | $70.90 \pm 2.39$ | $\mathbf{71.32} \pm 1.12$ | $70.00 \pm 0.73$ | $69.47 \pm 0.52$ | $70.27 \pm 2.04$ | $70.20 \pm 0.75$ |

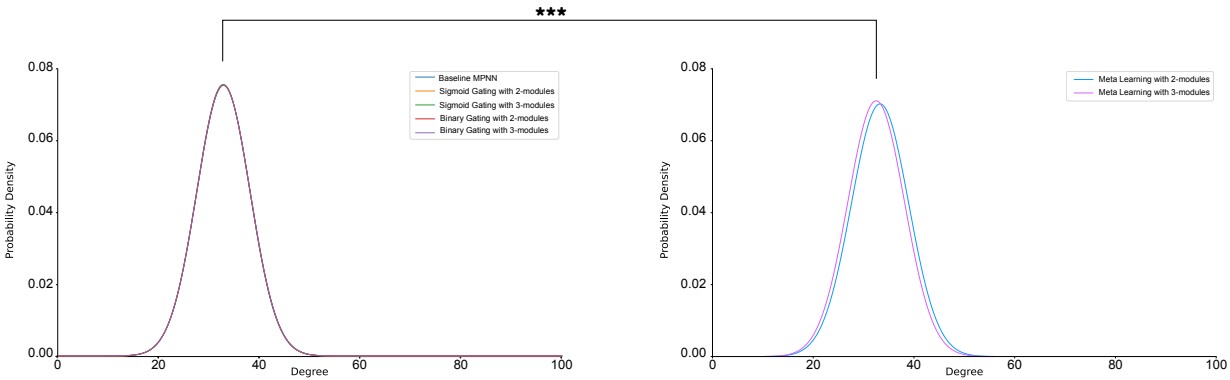

Figure A5: The degree distribution of high-performing test graphs when GNNs are trained on $\mathcal{G}_2$ is analyzed by fitting a Mixture of Gaussians. The resulting curves are presented in varying colors corresponding to the type of GNNs used. The curves on the left indicate rejection of the null hypothesis at a 5% significance level according to two-sample $F$-test. On the right, it is shown that the variance of the node degree distributions obtained by meta-learning-based GNNs differs from the baseline.

Table A17: $p$-value and test decision for two-sample $F$-test for each model configuration.

| Two-sample F-test | sigmoid gating (2 modules) | sigmoid-gating (3 modules) | binary-gating (2 modules) | binary-gating (3 modules) | meta-learning (2 modules) | meta-learning (3 modules) |
|---|---|---|---|---|---|---|
| $p$-value | 0.47 | 1.0 | 0.48 | 0.48 | 1.66e-29 | 6.69e-13 |
| $H_a : \sigma^2 = \sigma_0^2$ | True | True | True | True | **False** | **False** |

## A.8 Effect of module counts and few-shot OOD samples on generalization ability

All experiments in this paper were based on GNNs consisting of two update functions. To explore the impact of an additional update function and exposure to few-shot OOD samples, we trained the multi-module GNNs on $\mathcal{G}_2$ in graph theory multitask problem. Our results indicate that although the overall prediction accuracy remained unchanged, the generalization range could become wider. Consequently, we used a two-sample $F$-test to evaluate if there was a statistically significant difference in the variance of the average degree histogram between the single-module GNN baseline and the multi-module GNN histogram. Table A17 confirms that only the meta-learning-based GNN model increases the generalization scope, possibly because of the meta-testing phase's ability to reuse and restructure module assignment to adapt to a new environment. This finding suggests that multi-module GNNs with meta-learning are more advantageous when given access to a few instances from a novel environment, even if they do not match the OOD setting completely.

