# OpenReview forum: "Towards Better Generalization with Flexible Representation of Multi-Module Graph Neural Networks"
_TMLR — Accepted by TMLR_

### Review · Reviewer_JK6m · 2023-05-29

**Summary Of Contributions:**

This paper studies OOD generalization on graphs under graph size and structural distribution shifts. The authors present a random graph generator to study the effects of different graph properties to the OOD generalization of GNNs, and find the average node degree is the key factor. They propose multi module GNNs to handle multiple degree modes and provide certain experiments to show the effectiveness of their proposal.

**Audience:**

Yes

**Claims And Evidence:**

No

**Requested Changes:**

While I acknowledge the considerable effort put into the experiments, I believe there is room for improvement in the work, specifically in addressing the following weaknesses:

i) The discussions regarding the influence of average node degree and the Multimode degree distribution are limited:
- The phenomena that GNNs only generalize well to graphs that have similar degree distributions as the training data is straightforward and not interesting. It is believed that the other graph properties can also have the same effects. In other words, one could also count other graph properties such as graph size, clustering coefficients, and also find similar phenomena.

- Besides, the observation of average node degree is limited to synthetic datasets.

- Although the discussion regarding 5.2 seems to be interesting, it is unclear whether the OOD performance degeneration is caused by the shifts of average degrees or the existence of multimode degree distribution. And it is also unclear whether the same phenomenon exists in real world applications.

ii) The proposed multi-module network is not new. Some existing works such as PNA also adopts multiple aggregation functions. It is also unclear how multi module could resolve multimodal degree issue.

iii) The experiments are limited:
- First, in Figure 6, it seems none of the gating choices consistently outperform vanilla MPNN at all degree distributions. For example, in Figure 6e, the vanilla baseline performs better than the meta learning one at the largest degree.

- The performances of MPNN at Table 2 on GOOD benchmark have huge gaps than those reported by GOOD benchmark. The proposed multi-module method can not always outperform MPNNs.

- None of the previous OOD generalization methods are compared in experiments.



**Strengths And Weaknesses:**

**Strengths**

i) The experiments with the random graph generator are comprehensive.

ii) The figures are elegant and informative.


**Weakness**

i) The discussion regarding the influence of average node degree seems to be limited and uninteresting.

ii) The proposed multi-module network is not new.

iii) The experimental improvements are limited and no experimental comparison with existing OOD methods on graphs.

---

> ### Author Response · Authors · 2023-06-04
> **Response to Reviewer JK6m**
>
> Thank you for your thoughtful comments and suggestions. We appreciate your feedback and address your comments below:
>
> > Q1-1: Other graph properties can have the same effects.
>
> - Our extensive experiments on random graphs aims to uncover a generalization pattern of GNNs without imposing strict assumptions about the distribution of node degrees. We consider various graph properties, including clustering coefficient and path length, in addition to node degrees, and treat them equally when evaluating the generalization capability of GNNs. The generalization pattern, which strongly correlates with the average node degree, emerges from the set of diverse graph measures rather than being expected. Our experimental findings demonstrate that other graph properties do NOT exhibit similar correlations with the generalization ability of GNNs. Let us illustrate the rationale behind this observation.
>
> -  Suppose we imagine a hypothetical scenario where both the average and maximum degree influence the model's generalization, and we train the model on graph $\mathcal{G}_3$. As shown in Figure 2c, $\mathcal{G}_3$ is located in the central region of the top area in our low-dimensional space. According to Figure 2b, graphs with the same average or maximum degree as $\mathcal{G}_3$ are positioned along the white vertical or slanted bands, respectively. Based on this observation, we can anticipate that the generalization pattern will likely converge around $\mathcal{G}_3$ since the only point where the two white bands intersect is in close proximity to $\mathcal{G}_3$. If the clustering coefficient were the main factor influencing generalization, we would not observe a pattern similar to that shown in Figure 4. In this way, Figure 2b can serve as a useful tool for understanding which graph properties are relevant to the model's generalization.
>
> > Q1-2: Observation of average node degree is limited to synthetic datasets.
>
> - We agree that if we observe the same generalization pattern in real-world graphs, it would potentially strengthen the evidence for the impact of degree distribution on GNNs' generalization. However, we would like to draw attention to Figure 2d, where it is evident that most of these datasets sparsely populate or occupy only a small portion of the graph space. This implies that real-world graphs have limitations in representing the full range of graph properties. Our primary goal is to introduce a graph benchmark that encompasses a wide spectrum of graph properties and offers a standardized evaluation framework. To achieve this, it is crucial to have a benchmark that adequately covers the range of graph measures. Unfortunately, real-world graphs often fail to meet these criteria, making it more challenging to identify generalization patterns under the incomplete data distribution compared to synthetic graphs.
>
> -  While we recognize the importance of exploring the same evidence in realistic scenarios, we believe that our proposed benchmark can complement existing realistic graph benchmarks in exploring distribution shifts concerning various graph properties. We hope this explanation addresses your concerns and highlights the significance of our benchmark proposal and the comprehensive experiments.
>
> > Q1-3: Unclear cause of OOD performance degeneration.
>
> - Thank you for bringing this to our attention. We have carefully considered your question and conducted an analysis on the population-level test loss, as depicted in Figure 6(b,e), for baseline GNNs trained on different graph datasets, namely $\mathcal{G}_2$, $\mathcal{G}_3$, $\mathcal{G}_4$, and {$\mathcal{G}_2, \mathcal{G}_4$}.
> - Our analysis involves determining the minimum loss within the loss landscapes and measuring the average rate of change in these minimum losses, denoted as $\delta_A=\Delta \log_{10}D_{KL}$, during the transition from $\mathcal{G}_2$ to $\mathcal{G}_3$ and from $\mathcal{G}_3$ to $\mathcal{G}_4$, which corresponds to shifts in average degrees. Similarly, we measure $\delta_B$ when the training sets changed from $\mathcal{G}_2$ to {$\mathcal{G}_2, \mathcal{G}_4$} and from $\mathcal{G}_4$ to {$\mathcal{G}_2, \mathcal{G}_4$}, reflecting multi-mode degree distributions. The results show that the average increase in loss due to the shift in node degrees (with an average value of $\delta_A=0.16$) is smaller than the increase caused by the presence of multi-modal degree distributions (with an average value of $\delta_B=0.22$). This finding indicates that the decline in OOD performance can be primarily attributed to the existence of multi-modal degrees. It is worth noting that conducting this type of analysis on real-world graphs is not straightforward due to their limited coverage in graph space, as mentioned in the response to the previous question.

---

> > ### Comment · Reviewer_JK6m · 2023-06-05
> > **Response to authors**
> >
> > Thank you for your detailed explanation, which resolves my concern regarding Q1. I believe complementing the explanation above could make the discussion with respect to how and why avg node degree affects OOD generalization more informative. I am looking forward to your further response to my other questions.

---

> ### Author Response · Authors · 2023-06-04
> **Response to Reviewer JK6m**
>
> > Q2: PNA also adopts multiple aggregation functions. Unclear how multi module could resolve multimodal degree issue.
>
> - Thank you for raising this question. We aim to highlight two key distinctions between our proposed model and PNA. Firstly, while PNA focuses on determining how to combine incoming messages, our model emphasizes the strategy for performing nonlinear transformations on the combined message. In other words, PNA utilizes multiple aggregators, whereas our model employs multiple update functions. Secondly, PNA's multiple aggregators rely on predetermined mathematical operators, whereas our model selects or utilizes multiple update functions through learning processes. These differences set our model apart from PNA, as we prioritize distinct aspects of message processing and employ a learning-based approach for selecting update modules.
> - We appreciate your inquiry about the implications of the proposed multi-module framework in addressing the issue of multimodal degrees. In Figure 5(c,d), we have observed that multi-module GNNs tend to maintain or sometimes even reduce the complexity of the message distribution. In scenarios involving multimodal degrees, a single update function faces the challenge of handling statistically diverse message inputs aggregated from neighboring nodes with varying degrees. By employing multiple update modules and allowing them to selectively process messages through learning, it becomes possible to compensate for the shift in message statistics within a node by switching to a different type of update function. Consequently, the multi-module framework enables homogeneity in node representation and message vectors throughout the message-passing iterations. This interpretation could be further validated through additional theoretical and quantitative investigations. We acknowledge the necessity for such studies and have plans to explore them in future research endeavors.

---

> ### Author Response · Authors · 2023-06-04
> **Response to Reviewer JK6m**
>
> > Q3-1: None of the gating choices seems consistently outperform vanilla MPNN at all degree distributions.
>
> - We appreciate your valuable feedback. In relation to your comment regarding the inconsistent outperformance across all degrees, we would like to clarify the objective of our multi-module framework. Our primary goal was not to achieve consistent outperformance across all degrees, but rather to enhance generalization over graph properties beyond average degree. This is why we observe an improvement in the OOD performance along vertical patterns. According to the literature on OOD generalization, the mode of degree distribution is categorized as an environmental variable that can lead to difficulties in generalization, while graph size and other graph measures, excluding the average degree, are considered invariant features. Therefore, our focus was on improving generalization with respect to these invariant features, rather than aiming for uniform performance across multiple environments. Although we included the explanation in Section 6, we agree that it was not thorough, and will make sure to include all necessary clarifications in our future revisions.
>
> > Q3-2: MPNN at Table 2 on GOOD benchmark have huge gaps than those reported by GOOD benchmark. The proposed multi-module method can not always outperform MPNNs.
>
> - We apologize for any confusion and would like to provide clarification regarding the choice of baseline GNN in the original GOOD benchmark paper. It is important to note that the baseline GNN used in previous study was either GCN or GIN. However, for our research, we specifically selected MPNN as our baseline GNN. The reason behind this choice is that the update function in MPNN is defined as one of the fundamental mathematical operations necessary for the message-passing process. In contrast, vanilla GCN or GIN does not explicitly incorporate the update module in the same manner. This distinction influenced our decision to use MPNN as the baseline for our study. Therefore, it is not appropriate to directly compare the performance of MPNN in our study with the experimental results presented in the GOOD paper.
> - Furthermore, we would like to emphasize once again that the real-world graphs employed in the GOOD benchmark exhibit a relatively low degree and cover only a small fraction of the graph space (Figure 2d). This limited coverage poses challenges when attempting to conduct similar experiments on real graphs within a controlled environment, as we have done with synthetic data. Consequently, our multi-module framework, which enhances generalization with respect to aforementioned invariant features, cannot be expected to consistently outperform the baseline, considering the fact that real-world graphs rarely conform to our hypothesis of multi-mode degree distribution. In this regard, the number of tasks in which our proposed framework outperforms the baseline (19 out of 27 wins) is a rather surprising outcome and warrants further investigation in future research.
>
> > Q3-3: None of the previous OOD generalization methods are compared in experiments.
>
> - Based on the same underlying rationale discussed in the previous response, our main focus in Table 2 was to compare the proposed model with the single module GNN baseline. Comparing our model with other OOD generalization methods on real data, which do not readily conform to our initial hypothesis, was deemed unnecessary in the absence of a clear interpretation method. Nevertheless, we are currently experimenting with previous OOD generalization methods and will share our findings as they unfold.

---

> ### Author Response · Authors · 2023-06-29
> **Response to Reviewer JK6m**
>
> > Q3-3 (continued): None of the previous OOD generalization methods are compared in experiments.
>
> - We have completed the additional experiments as requested and would like to share our findings. Firstly, we conducted real-world graph experiment once again by comparing the single-module baseline with the multi-module framework. We used two different aggregators (max or sum) across all tasks and reported the best outcome achieved among these two aggregations. By doing so, we aimed to minimize the performance gaps between our approach and the results reported by GOOD. Even in the new experiments, the number of tasks in which the proposed framework outperformed the baseline remained unchanged (see Table 2).
> - Secondly, we have included the results of two well-established OOD generalization methods, namely invariant risk minimization (IRM) [1] and variance risk extrapolation (VREx) [2], in Table A6. In this experiment, the baseline, multi-module GNNs, and existing OOD generalization methods performed the best on 6, 13, and 8 tasks, respectively. Although these additional experiments yield mixed results, they continue to provide encouragement by demonstrating the predictive power of multi-module GNNs in specific tasks.
>
> ​[1] Martin Arjovsky, Léon Bottou, Ishaan Gulrajani, and David Lopez-Paz. Invariant risk minimization. *arXiv preprint arXiv:1907.02893*, 2019.
>
> [2] DavidKrueger,EthanCaballero,Joern-HenrikJacobsen,AmyZhang,JonathanBinas,Dinghuai Zhang, Remi Le Priol, and Aaron Courville. Out-of-distribution generalization via risk extrapolation (REx). In *International Conference on Machine Learning*, pages 5815–5826. PMLR, 2021.

---

### Review · Reviewer_QihW · 2023-06-10

**Summary Of Contributions:**

The authors inspect the behavior of GNNs under out of distribution (OOD) settings, focusing on the generalization ability of GNNs.

To do so, the authors propose two main steps:
1. Generate a large random graph dataset with varying parameters, such that graphs with a large range of mean degree are obtained and with different motifs. The authors use this dataset to evaluate the generalization of GNNs.

2. The authors propose a multi-module GNN framework that is supposed to capture different node feature aggregation rules, based on their hypothesis and observation that in cases where underlying graph properties may vary between train and inference stages (e.g., the mean node degree), using a single aggregation rule results in worse performance.

The authors evaluate their method on several synthetic and real-world datasets, where they compare the performance of different variants of their method with a standard MPNN.


**Audience:**

Yes

**Claims And Evidence:**

Yes

**Requested Changes:**

Please see my questions in the weaknesses section in my review.

**Strengths And Weaknesses:**

Strengths:
1. The paper is well motivated
2. The authors propose both qualitative (i.e., analyses), quantitative (empirical comparison), and software/date (the proposed framework and dataset) contributions.
3. The results show the importance and empirical contribution of using several node aggregation rules and the influence of the mean node degree on performance.

Weaknesses:

1. The experimental comparisons are lacking. While comparing the proposed method to a baseline MPNN, it is also essential to compare it with other OOD GNN works.
2. The authors propose to use multiple aggregation rules. It seems to me that the authors focus on the activation part of the network. I would like to ask the authors if they can relate to the use of various (whether pre defined or learned) node propagation rules. Some examples to such ideas are present in [1],[2],[3],[4].
3. The authors widely discuss the effect of different motifs and underlying structures found in the input graphs and their influence on the generalization of GNNs. In this context, can the authors elaborate on the relation to recent positional/structural encoding methods? (PE/SE). Some examples are [5],[6].

[1] Complete the missing half: Augmenting aggregation filtering with diversification for graph convolutional networks

[2] Revisiting heterophily for graph neural networks

[3] Adaptive Universal Generalized PageRank Graph Neural Network

[4] pathGCN: Learning General Graph Spatial Operators from Paths

[5] GRAPH NEURAL NETWORKS WITH LEARNABLE STRUCTURAL AND POSITIONAL REPRESENTATIONS

[6] Improving Graph Neural Network Expressivity via Subgraph Isomorphism Counting

---

> ### Author Response · Authors · 2023-06-19
> **Response to Reviewer QihW**
>
> Thank you for the encouraging comments and valuable suggestions. We appreciate your feedback and address your comments below:
>
> > Q1: It is also essential to compare the proposed method with other OOD GNN works.
>
> - In response to your and reviewer JK6m's requests, we are currently conducting additional experiments on previous OOD generalization methods. We are actively working on these experiments and will promptly share the results once they become available.
>
> > Q2: I would like to ask the authors if they can relate to the use of various node propagation rules. Some examples to such ideas are present in [1],[2],[3],[4].
>
> - We appreciate your insightful references. You accurately highlight the distinctive features of our proposed method compared to [1-4]. To delve into more detail, [1] and [2] suggested employing multiple channel filter banks to enrich node representations. This concept aligns with the idea of utilizing multiple message functions within the MPNN framework to generate different types of incoming messages. In this case, the node representation is obtained by combining multiple aggregated messages through a weighted sum, followed by a simple activation function. Both [1-2] and our proposed method employ multiple nonlinear transformations; however, the dissimilarity lies in whether the nonlinear transformation is applied to the message function or the update function. Furthermore, [1-2] specifically focuses on improving node classification tasks on heterophilic graphs using a sigmoid gating strategy, whereas our approach avoids limiting itself to particular task types and explores various ways of leveraging multiple modules.
>
> - While previous approaches have primarily concentrated on methodologies within a single GNN layer, [3-4] have shifted their focus towards enhancing the quality of the final node representations by adaptively leveraging the outputs of each propagation step, rather than solely relying on the final layer output. We appreciate your input and will incorporate all the suggested references in our revised manuscript.
>
> > Q3: Can the authors elaborate on the relation to recent positional/structural encoding methods? (PE/SE). Some examples are [5],[6].
>
> - Thank you for bringing up the informative references. The utilization of positional/structural information encoding (PE/SE) has emerged as a prevalent strategy to overcome the limited expressivity of MPNNs. In this approach, each node explores the global structure of the input graphs through multiple rounds of message passing. However, this method is prone to a well-known over-smoothing issue. To address this concern, [5] and [6] introduce a solution by integrating positional/structural features into the canonical message-passing operation, enabling simultaneous updates of both node and positional/structural representations. Our proposed approach is comparable to [5-6] in the sense that the multi-module framework incorporates additional update modules instead of additional input features. This allows for exponentially many combinations of nonlinear transformations, enabling the adjustment of representations to handle various structures present in the input graphs. Although our framework does not aim to directly capture local sub-structures of the underlying graph as in [5-6], we will consider the effectiveness of our model for this purpose in future research.
>
> - Additionally, in our study, we have also applied PE in the context of graph theory multi-task and CLRS algorithmic reasoning task. Notably, even in these experiments, our proposed multi-module framework outperforms the baseline with PE. This indicates that the combination of the multi-module framework and PE/SE mutually complement each other in enhancing the generalization of GNNs. We sincerely appreciate your valuable input, and we will ensure to incorporate all the suggested references into our revised manuscript.

---

> ### Author Response · Authors · 2023-06-29
> **Response to Reviewer QihW**
>
> > Q1 (continued): It is also essential to compare the proposed method with other OOD GNN works.
>
> - The additional experiments requested have been successfully completed. For further details, please refer to our latest response addressed to reviewer JK6m.

---

### Review · Reviewer_9i1U · 2023-06-18

**Summary Of Contributions:**

In this work, the authors investigate how structural properties of graphs affect the OOD generalization of GNNs. Using a novel random graph benchmark, they find that the average node degree is a crucial factor for scaling to OOD graphs. To overcome the performance drops caused by the distributional shift in this structural property, they design a multi-module MPNN framework that combines various update functions for node representations. The experimental results show that the proposed framework improves the predictive performance on most considered problems of graph learning.

**Audience:**

Yes

**Broader Impact Concerns:**

I do not have any concerns on the ethical implications of the work.

**Claims And Evidence:**

Yes

**Requested Changes:**

1. If I understand correctly, the assignment of yellow and blue colors in the caption of Figure 4 should be swapped;
2. If possible, I suggest to add a pseudocode (or more compact description) that clearly explains how the proposed meta-learning procedure works.

**Strengths And Weaknesses:**

## Strengths
1. According to Section 3.1 and Figure 2, a new random graph benchmark covers much more space compared to other established graph datasets (at least in terms of the structural graph characteristics discussed in this work). This approach to sample graphs might be particularly interesting for the research areas focused on the methods for generating unique graphs with diverse structural properties;
2. In Section 4, the authors propose various different methods to design multi-module MPNNs, including binary / sigmoid gating and meta-learning technique, that share the same motivation about how it may be useful to have different update functions in MPNN. This allows one to choose the architecture for their needs, which is especially useful since the most complex architectures do not always show the best performance, as shown in Section 5.4;
3. In Section 5.1, the proposed method of investigating the impact of each structural property by comparing the directions of changes in the structural properties of nodes with the direction of changes in the predictive performance on the constructed 2D plane seems clever and intuitive;
4. As shown in Section 5, the importance of average node degree of the input graphs is proved by consistent results of numerous experiments. In particular, the authors consider synthetic data from the proposed random graph generator and CLRS benchmark, as well as natural graphs from GOOD benchmark; the investigated problems include marginal inference, graph theory multi-task prediction, standard graph classification, and algorithmic reasoning.

## Weaknesses
1. In Section 3.2, it remains unclear how the proposed method for generating the input features on nodes and edges for making marginal inference and solving graph theory affects the main conclusions of the empirical study about the importance of average node degree. In other words, do these claims hold for any other approach to generate features on nodes and edges, and whether this question is important to discuss at all;
2. In Section 4.3, the description of a multi-module meta-learning is somewhat hard to follow, especially with regard to the data used and the combination of loss functions applied. As a suggestion, the authors may provide a pseudo-code for their meta-learning algorithm that clearly describes the alternating procedure to obtain the optimal assignment of update module;
3. As shown in Section 5.4, there may be notable variations in the performance of the proposed meta-learning framework across different prediction tasks on real-world datasets, and this does not allow one to draw clear conclusions as to whether it is really worth using a complex procedure for constructing update modules instead of simple methods like sigmoid or binary gating.
4. According to Figure A4, the structural properties of most real-world datasets do not reveal the same transition of average node degree across train, validation, test subsets as simulated by the proposed random graph benchmark. It means that there is no such distributional shift for which the multi-module MPNN architecture has been proposed. This fact may explain the inconsistent results with high variation on real-world datasets. For the sake of fairness, it should be noted that the authors discuss the limitations of the proposed random graph benchmark in Section 5.4. In particular, they mention that while it allows to cover a multi-modal distribution of graphs in terms of their structural properties, it does not necessarily reproduce the same properties as in real-world datasets.

---

> ### Author Response · Authors · 2023-06-29
> **Response to Reviewer 9i1U**
>
> We thank the reviewer for their support and insightful thoughts. We appreciate your feedback and address your comments below:
>
> > Q1: In Section 3.2, it remains unclear how the proposed method for generating the input features affects the main conclusions about the importance of average node degree.
>
> - We thank the reviewer for raising this question. While the primary focus of our research is on the generalization performance of GNNs under structural distribution shifts, we acknowledge that the results obtained from synthetic experiments may lead to inquiries regarding the applicability of our findings to different input features. Specifically, in the marginal inference task, we sampled node and edge features from a standard normal distribution, denoted as $\mathcal{N}(0,1)$, whereas the graph theory task involved sampling from a uniform distribution $\mathcal{U}[0,1]$. If our objective were to explore the implications of changing the distribution of input features, such as transitioning from $\mathcal{N}(0,1)$ to $\mathcal{N}(0,1/4)$ across the entire sample space, our previously conducted experiments (which were not reported) demonstrated a consistent generalization pattern with the results presented in this study, as long as such a change did not adversely impact the predictive performance of GNNs. However, if the question pertains to the same change in the distribution of input features between training and test graphs, it may not be guaranteed that our claims will remain valid. This is because the generalization of GNNs relies heavily on the interaction between graph structure and features. Nevertheless, we believe our study holds significance as we strive to isolate and examine the impact of structural distribution shifts on the generalization ability of GNNs, an investigation that has not been undertaken previously.
>
> > Q2: As a suggestion, the authors may provide a pseudo-code for their meta-learning algorithm.
>
> - We appreciate your valuable suggestion. In Appendix 4, we have included a PyTorch code snippet of the multi-module meta-learning algorithm, accompanied by line-by-line comments for better understanding.
>
> > Q3: As shown in Section 5.4, there may be notable variations in the performance ...
> >
> > Q4: According to Figure A4, the structural properties of most real-world datasets do not reveal the same transition ...
>
> - Thank you for sharing your feedback with us. Your observation is absolutely correct, and we want to reiterate that the real-world graphs used in the GOOD benchmark do not display a clear shift in structural distribution that aligns with the discussed scenario in synthetic data. Moreover, the OOD tasks provided in GOOD benchmark are not specifically designed to study the structural OOD in graph domains. This discrepancy poses challenges when trying to conduct similar experiments on real graphs within a controlled environment, as we have done with synthetic data. As a result, our multi-module framework, which enhances generalization based on certain invariant features, cannot be expected to show a consistent outperformance among many other OOD generalization methods and even within our own strategies. Despite these limitations, we believe our study remain significant because current real graph benchmarks do not necessarily represent the entire spectrum of graph structures.
> - Furthermore, in many engineering and scientific applications, the focus often lies on designing an optimal graph structure rather than processing a given graph. For example, the graph structure in error-correcting code (ECC) algorithms plays a crucial role in determining the code's error correction capabilities, decoding complexity, and convergence behavior. Optimized graph structures are crafted to achieve favorable trade-offs between performance and complexity, enabling efficient and reliable error correction in practical applications. From this perspective, our experimental results can complement the understanding of network-based ECCs' behavior and aid in designing more efficient neural inference models, which will be the focal point of our upcoming research endeavors.
>
> > Q5: If I understand correctly, the assignment of yellow and blue colors in the caption of Figure 4 should be swapped.
>
> - Thank you for pointing that out, and we apologize for any confusion it may have caused. Upon reviewing the caption and Figure 4 again, We can confirm that the assignment of yellow and blue colors in the caption is correct as presented. If you have any further questions or concerns, please let us know, and we will be happy to assist you.

---

### Review · Reviewer_8Lyz · 2023-06-21

**Summary Of Contributions:**

This paper is concerned with how the graph structure influences the generalization of GNNs. OOD of GNNs is considered and is verified to have high correlation with the average degree of graphs by the random graph generators.

**Audience:**

Yes

**Claims And Evidence:**

Yes

**Requested Changes:**

1.	Will other topological properties (such as higher-order topology from the populated graph) of graph besides the notion from network science affect the OOD and predictive performance of the GNNs and how?
2.	What is the benefit of using Gumbel reparameterization trick?
3.	Despite the empirical study, is there any theoretical justification and guarantee for the performance improvement of introducing multi-module meta-learning?

**Strengths And Weaknesses:**

The papers combine multiple update modules and meta-learning method for neural message passing to solve the challenge when multi-mode degree distribution is involved. The performance of the proposed model is verified by both synthetic and real-world data.

---

> ### Author Response · Authors · 2023-06-29
> **Response to Reviewer 8Lyz**
>
> We appreciate your thorough review of our paper and the valuable feedback you have provided.
>
> > Q1: Will other topological properties affect the OOD and predictive performance of the GNNs and how?
>
> - That's an interesting question. The influence of topological properties beyond traditional network science notions on the OOD and predictive performance of GNNs, is an area that extends beyond the scope of our current studies. While it is conceivable that higher-order topology could impact GNN performance, further investigation and research would be necessary to fully understand the extent of its effects and the underlying mechanisms involved. It is an intriguing direction for future research that warrants further exploration and study.
>
> > Q2: What is the benefit of using Gumbel reparameterization trick?
>
> - We appreciate your question. Initially, our proposal of the binary gating with Gumbel reparameterization trick was not driven by any specific anticipated benefits. Rather, it was chosen as the simplest strategy to test in scenarios where each node needed to select one from multiple update modules. The binary gating strategy can be considered as sitting in the middle of the spectrum among the three strategies we introduced. During the forward pass, binary gating employs discrete sampling similar to multi-module meta-learning, while during the backward pass, computations are carried out on a computational graph defined by multiple modules, akin to sigmoid gating. We acknowledge the importance of this clarification and will incorporate it into a revised manuscript.
>
> > Q3: Is there any theoretical justification and guarantee for the performance improvement of introducing multi-module meta-learning?
>
> - Thank you for your insightful question. While our empirical study provides evidence of the performance improvement achieved by introducing multi-module meta-learning, it is important to note that our current research has primarily focused on the practical implementation and evaluation of this approach. At this stage, we may not have a comprehensive theoretical justification or guarantees for its performance improvement. However, we appreciate your interest in further theoretical investigations, and our study can indeed serve as a valuable starting point for such endeavors.

---

### Author Response · Authors · 2023-06-29
**General Response to Reviewers and Action Editors**

We would like to extend our appreciation to the reviewers for their invaluable feedback and numerous constructive suggestions. We are encouraged by their positive assessment of our paper, recognizing it as well-motivated and comprehensive (R1: JK6m), featuring useful analysis and empirical comparisons (R2: QihW), clearly presenting the contributions (R2: QihW), offering an informative approach to investigate the impact of structural properties on OOD generalization (R3: 9i1U), and its validation of the proposed model using both synthetic and real-world data (R4: 8Lyz).

We have incorporated the feedback provided by the reviewers into our updated manuscript. Here is a summary of the changes made:

- Provided further clarification in Section 6 to better explain how our empirical results can be interpreted within the existing literature on OOD generalization in graphs.
- Expanded the related works section in Section 6 to include additional references pertaining to the utilization of various node propagation rules.
- Included the results of existing OOD generalization methods in Section 5.4 and Table A6.
- Added the pseudocode of multi-module meta-learning in Appendix 4.
- Clarified the interpretation of the "binary gating mechanism" in relation to the three different strategies employed for leveraging multiple update modules within our framework.
- Made minor clarifications throughout the text as suggested by the reviewers.

If you require further clarification on any specific aspect, please let us know.

Sincerely,

Authors

---

### Decision · Action_Editors · 2023-07-19

**Recommendation:** Accept as is

**Comment:**

The submission tackles the important and timely problem of OOD generalisation of graph neural networks. The authors have performed several thorough controlled experiments on synthetic graphs and have exploited the conclusions to design a more generalisable model. Three reviewers are positive about the submission, and I also recommend acceptance.

**Audience:**

Yes, in my opinion, the results in the submission will be interesting to a large part of the GNN community.

**Claims And Evidence:**

The authors have identified the dependency of the GNNs OOD generalization ability and the average node degree of the input graphs. While this dependency was observed on the synthetic graphs, this evidence allowed to design a new model that was shown to outperform the competitors on several real-world graphs. Therefore, the claims are well-supported.